# VisionTS: Visual Masked Autoencoders Are Free-Lunch Zero-Shot Time Series Forecasters

## Abstract

Foundation models have emerged as a promising approach in time series forecasting (TSF). Existing approaches either repurpose large language models (LLMs) or build large-scale time series datasets to develop TSF foundation models for universal forecasting. However, these methods face challenges due to the severe cross-domain gap or in-domain heterogeneity. This paper explores a new road to building a TSF foundation model from rich, high-quality natural images. Our key insight is that a visual masked autoencoder, pre-trained on the ImageNet dataset, can naturally be a numeric series forecaster. By reformulating TSF as an image reconstruction task, we bridge the gap between image pre-training and TSF downstream tasks. Surprisingly, without further adaptation in the time-series domain, the proposed VisionTS could achieve superior zero-shot forecasting performance compared to existing TSF foundation models. With fine-tuning for one epoch, VisionTS could further improve the forecasting and achieve state-of-the-art performance in most cases. Extensive experiments reveal intrinsic similarities between images and real-world time series, suggesting visual models may offer a "free lunch" for TSF and highlight the potential for future cross-modality research. Our code is available in the Supplementary Material.

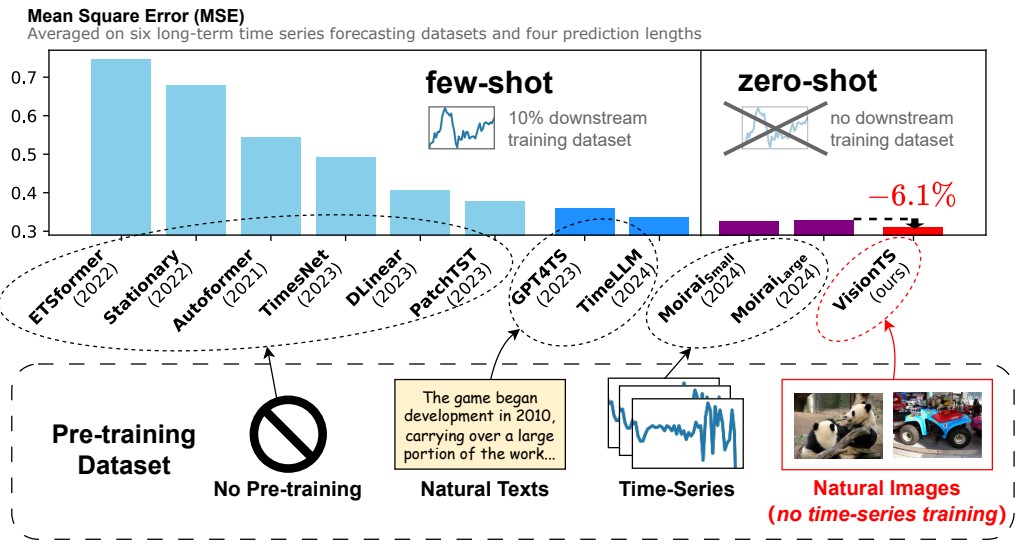

Figure 1: Long-term forecasting performance comparison. Our VisionTS, *without any training on time series data*, outperforms the largest foundation model MoiraiLarge in the zero-shot setting.

## 1 Introduction

Foundation models (Bommasani et al., 2021) have revolutionized natural language processing (NLP) and computer vision (CV) in recent years (Brown et al., 2020; He et al., 2022). By pretraining on large-scale data, they have shown remarkable few-shot and even zero-shot performance across various

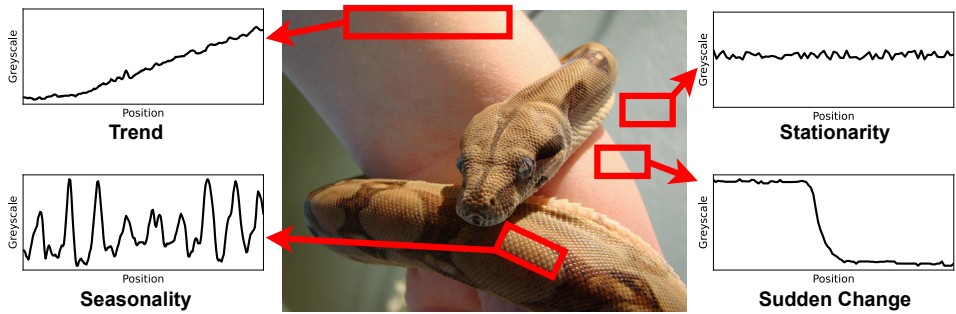

Figure 2: An image of the ImageNet dataset (Deng et al., 2009), in which the pixel arrays can display many well-known features of real-world time series, such as trend, seasonality, and stationarity (Qiu et al., 2024). By self-supervised pre-training on ImageNet, it is reasonable that a visual model could understand these features and exhibit a level of time series forecasting ability.

downstream tasks. This has motivated an emergent paradigm shift in time series forecasting (TSF), moving from a traditional one-model-per-dataset framework to *universal forecasting* with a single pre-trained model (Woo et al., 2024; Goswami et al., 2024). A TSF foundation model can greatly reduce the need for downstream data and demonstrate strong forecasting performance on diverse domains, such as energy consumption planning, weather forecasting, and traffic flow.

We have recently witnessed two roads to building a TSF foundation model for universal forecasting. The *first* tries to repurpose large language models (LLMs) that have been pre-trained on text data for TSF tasks (*i.e.*, **text-based**) (Zhou et al., 2023; Jin et al., 2024), based on the observation that LLMs and TSF models share a similar left-to-right forecasting paradigm. However, due to the significant gap between these two modalities, the effectiveness of such transferability between language and time series has recently been questioned by Tan et al. (2024).

The *second* road focuses on constructing large-scale time-series datasets collected from diverse domains to train a TSF foundation model from scratch (*i.e.*, time series-based or **TS-based**) (Woo et al., 2024; Das et al., 2024). Nevertheless, unlike images or language with unified formats, time series data is highly heterogeneous in length, frequency, number of variates, domains, and semantics, limiting the transferability between pre-training and downstream domains. Until recently, constructing a high-quality dataset remains challenging and is still in the early exploration stage.

In this paper, we investigate a *third* road that is less explored yet promising: building TSF foundation models with pre-trained *visual* models. Our key idea is that pixel variations in a natural image can be interpreted as temporal sequences, which share many intrinsic similarities with time series: ❶ **Similar modalities**: Unlike discrete texts, both images and time series are continuous; ❷ **Similar origin**: Both time series and images are observations of real-world physical systems, whereas languages are products of human cognitive processes; ❸ **Similar information density**: Languages are human-generated signals with high semantic density, while images and time series are natural signals with heavy redundancy (He et al., 2022); and ❹ **Similar features**: As shown in Fig. 2, images often display many features of real-world time series, which are rarely found in language data. Based on these findings, images could be a more promising modality for transferring to TSF. We are motivated to answer the question: *Can a visual model pre-trained on images be a free-lunch foundation model for zero-shot time series forecasting?*

We focus on visual masked autoencoder (MAE)[1], a popular CV foundation model (He et al., 2022) by self-supervised pre-training on ImageNet (Deng et al., 2009). As an image reconstruction and completion model, MAE can naturally be a *numeric series forecaster*. Inspired by the well-known prompt technique in NLP (Schick & Schütze, 2021), we propose a simple method to reformulate TSF as a patch-level image reconstruction task to bridge the gap between pre-training and downstream tasks. Specifically, we transform 1D time series data into 2D matrices via segmentation. Then, we render the matrices into images and align the forecasting window with masked image patches. This method allows us to make zero-shot forecasting without further adaptation.

---

[1] We use fonts to distinguish MAE (Masked Autoencoder) and MAE (Mean Absolute Error) in this paper.

We evaluate our proposed VISIONTS on 43 TSF benchmarks across various domains, including long-term TSF datasets (Zhou et al., 2021; Wu et al., 2021), Monash (Godahewa et al., 2021), and PF (Woo et al., 2024). As demonstrated in Fig. 1, *without* any further adaptation in the time-series domain, a vanilla `MAE` can surprisingly achieve a comparable performance or even outperform the state-of-the-art (SOTA) zero-shot TSF foundation models, including text-based and TS-based methods. By fine-tuning `MAE` on each downstream dataset for only one epoch, VISIONTS can lead to a SOTA performance on most long-term TSF benchmarks.

To further understand and explain the transferability, We use an `MAE` encoder to visualize both modalities, showing a level of similarity between time series and natural image representations. Additionally, we observe considerable heterogeneity within time series data, and images can serve as a bridge to connect these isolated time series representations. Our findings suggest that time series and natural images may be two sides of a coin, and visual models can be a free lunch for time series forecasting. We hope our findings can inspire future cross-modality research on CV and TSF.

Our contributions are summarized as follows:

- We explore a road to building a TSF foundation model from natural images, which is conceptually different from the existing text-based and TS-based pre-training methods.

- We introduce VISIONTS, a novel TSF foundation model based on a visual `MAE`. To bridge the gap between the two modalities, we reformulate the TSF task into an image reconstruction task.

- Comprehensive evaluations of VISIONTS on 43 benchmarks across multiple domains demonstrate its significant zero-shot forecasting performance, surpassing few-shot text-based TSF foundation models and achieving comparable or superior results to zero-shot TS-based models.

## 2 PRELIMINARIES

**Time Series Forecasting (TSF)**  For a multivariate time series with $M$ variables, let $\boldsymbol{x}_t \in \mathbb{R}^M$ represent the value at $t$-th time step. Given a historical sequence (*i.e.*, look-back window) $\boldsymbol{X}_{t-L:t} = [\boldsymbol{x}_{t-L}, \cdots, \boldsymbol{x}_{t-1}] \in \mathbb{R}^{L \times M}$ with context length $L$, the TSF task is to predict future values (*i.e.*, forecast horizon) with prediction length $H$: $\hat{\boldsymbol{X}}_{t:t+H} = [\boldsymbol{x}_t, \cdots, \boldsymbol{x}_{t+H-1}] \in \mathbb{R}^{H \times M}$.

**Patch-Level Image Reconstruction**  To obtain high-quality visual representation for downstream CV tasks, He et al. (2022) proposed masked autoencoder (`MAE`) to pre-train a Vision Transformer (ViT) (Dosovitskiy et al., 2021) using a patch-level image reconstruction task on ImageNet. Specifically, for an image of size $W \times W$ (where $W$ represents both the width and height, as ImageNet images are square), the image is evenly divided into $N \times N$ patches, each with a width and height of $S = W/N$. During pre-training, some random patches are masked, while the remaining *visible patches* are fed into the ViT with their position encodings. `MAE` are trained to reconstruct the masked pixel values from these visible patches.

## 3 METHODOLOGY

As noted in the Introduction, time series and images share intrinsic similarities, suggesting the transfer potential of pre-trained visual models (particularly `MAE` in this paper) for TSF tasks. We explain how to reformulate TSF tasks into `MAE`'s pre-training task, *i.e.*, patch-level image reconstruction.

Our high-level idea is straightforward: map the look-back/forecasting windows to visible/masked patches, respectively. This idea is supported by the recent success of prompt tuning (Schick & Schütze, 2021) in NLP, where the predictions for `[mask]` token in pre-trained language models, *e.g.*, BERT (Devlin et al., 2019), are directly used for downstream tasks. By unifying the forms of the two tasks, we bridge the gap between the two domains, enabling a `MAE` for *zero-shot* TSF directly without adapting the pre-trained parameters.

Notably, this idea is limited to univariate forecasting since multivariates are intractable to be encoded in a single image. Fortunately, recent work shows that channel independence — predicting each variable separately for multivariate forecasting — can be highly effective (Nie et al., 2022; Han et al., 2024). Therefore, we leave the exploration of multivariate interactions for future work.

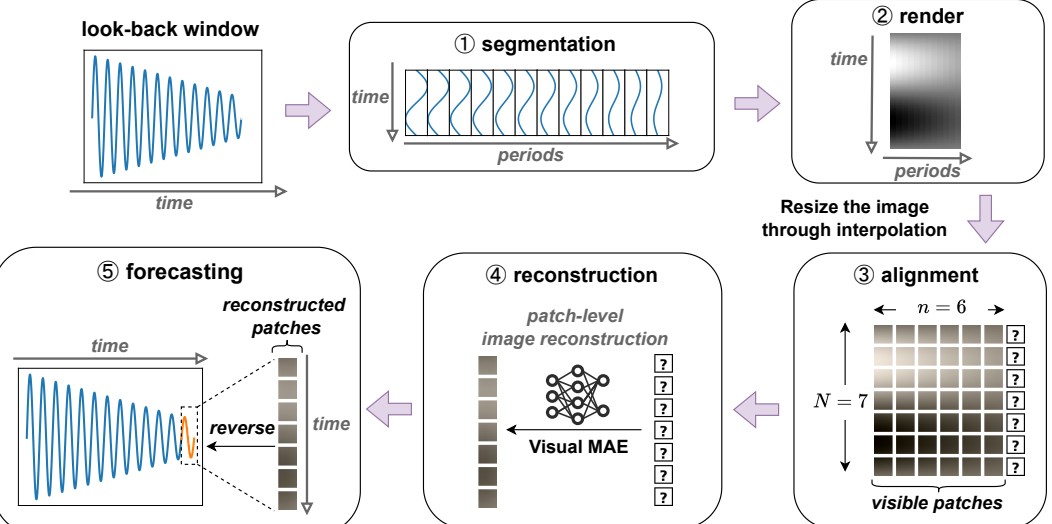

Figure 3: VISIONTS architecture. The input is first segmented by period, rendered into a grayscale image, and then aligned with the visible patches on the left through resampling. MAE is used to predict the masked patches on the right, and the reconstructed image is then reversed to forecasting.

However, implementing this idea poses a challenge: the dimension of time-series data (1D) is different from images (2D). Moreover, the size of images in the pre-training dataset is fixed at $224 \times 224$, while the lengths of time series data can vary dynamically. In the following, we describe the details of VISIONTS to address this challenge. Our architecture is depicted in Fig. 3.

**Segmentation** Given a univariate input $X \in \mathbb{R}^L$, the first goal is to transform it into a 2D matrix. We propose to segment it into $\lfloor L/P \rfloor$ subsequences of length $P$, where $P$ is the periodicity. Notably, when the time series lacks clear periodicity, we can set $P = 1$ directly, which is also effective in our experiments (Appendix B.4). In practice, $P$ can be determined using statistical methods like Fast Fourier Transform (Wu et al., 2023; Chen et al., 2024) or domain knowledge like sampling frequency (Godahewa et al., 2021; Alexandrov et al., 2020). In this paper, we select $P$ based on the sampling frequency, elaborated in Appendix A.2.

After that, these subsequences are then stacked into a 2D matrix, denoted by $\boldsymbol{I}_{\text{raw}} \in \mathbb{R}^{P \times \lfloor L/P \rfloor}$. This encoding strategy is proven to be efficient by recent work like TimesNet (Wu et al., 2023) and SparseTSF (Lin et al., 2024), as it allows for the simultaneous capture of both variations within the same period (*i.e.*, intra-period) and across periods with the same phase (*i.e.*, inter-period). Moreover, it ensures that each element in $\boldsymbol{I}_{\text{raw}}$ and its neighbors align with the *spatial locality* property of images (Krizhevsky et al., 2012), where nearby pixels tend to be similar due to the inherent cohesiveness of objects in the real world. Therefore, this further narrows the gap between time series and images.

**Normalization** MAE standardizes each image based on the mean and standard deviation computed on ImageNet. Therefore, we apply instance normalization to $\boldsymbol{I}_{\text{raw}}$, which is also a standard practice in current TSF (Kim et al., 2022). Notably, we observed that normalizing $\boldsymbol{I}_{\text{raw}}$ to a standard deviation of $r$, where $r$ is a hyperparameter less than 1, yields superior performance. One explanation is that the magnitude of inputs/outputs during MAE pretraining is constrained by the limited range of color values. Therefore, reducing the magnitude of $\boldsymbol{I}_{\text{raw}}$ prevents exceeding these limits. However, an excessively low $r$ can result in values that are difficult to distinguish. We found that a moderate value (0.4) of $r$ performs well across most scenarios (See Appendix B.8 for more details). Let $\boldsymbol{I}_{\text{norm}}$ denote the normalized matrix, which is computed as follows:

$$\boldsymbol{I}_{\text{norm}} = r \cdot \frac{\boldsymbol{I}_{\text{raw}} - \text{Mean}(\boldsymbol{I}_{\text{raw}})}{\text{Standard-Deviation}(\boldsymbol{I}_{\text{raw}})}.$$

**Rendering** It is well-known that each image has three channels. We simply render $\boldsymbol{I}_{\text{norm}}$ as a grayscale image $\boldsymbol{I}_{\text{grey}} \in \mathbb{R}^{P \times \lfloor L/P \rfloor \times 3}$, where all three channels are identical to $\boldsymbol{I}_{\text{norm}}$. This choice

is purely result-driven: In our early experiments, we added a convolutional layer with three output channels to convert the grayscale image into a color image and then fine-tuned it to find the optimal color transformation, which, however, did not significantly influence the performance.

**Alignment** Our goal is to predict the columns on the right of $\boldsymbol{I}_{\text{grey}}$ to forecast the future sequence. A straightforward approach is to treat $\boldsymbol{I}_{\text{grey}}$ as the visible left portion and the predicted columns as the masked right portion. However, since the image size during pre-training may not match the size of $\boldsymbol{I}_{\text{grey}}$, we propose to resize $\boldsymbol{I}_{\text{grey}}$ to align with the pre-training data. Formally, let the total number of 2D patches used in pre-training be $N \times N$ and the size of each patch be $S \times S$. We set the number of visible patches to $N \times n$ and the masked patches to $N \times (N - n)$, where $n = \lfloor N \cdot {}^L/_{(L+H)} \rfloor$ is determined by the ratio of context length $L$ to prediction length $H$. We resample the image $\boldsymbol{I}_{\text{grey}}$ to adjust the size from the original dimensions $(P, \lfloor L/P \rfloor)$ to $(N \cdot S, n \cdot S)$, making it more compatible with MAE. We select *bilinear interpolation* for the resampling process.

Moreover, we found that reducing the width of the visible portion can further improve performance. One possible explanation is that MAE uses a large masked ratio during pre-training, with only 25% of patches visible. Reducing the image width may align the masked ratio more closely with pre-training. Therefore, we propose multiplying $n$ by a hyperparameter $c \in [0, 1]$. Similar to $r$, we found that setting $c = 0.4$ performs well in our experiments (See Appendix B.8). Final $n$ can be formulated as:

$$ n = \left\lfloor c \cdot N \cdot \frac{L}{L + H} \right\rfloor . $$

**Reconstruction and Forecasting** After obtaining the MAE-reconstructed image, we simply reverse the previous steps for forecasting. Specifically, we resize the entire image back to the original time series segmentations through the same bilinear interpolation, and average the three channels to obtain a single-channel image. After de-normalizing and flattening, the forecasting window can be extracted.

## 4 EXPERIMENTS

We use MAE (Base) as our backbone, while we also test other sizes of MAE and LaMa (Suvorov et al., 2022) afterward. We select representative baselines for comparison, including two **TS-based** foundation models, three **Text-based** foundation models, and **other popular TSF baselines** covering both Transformer-based, MLP-based and CNN-based architectures. Baseline and benchmark details are elaborated in Appendix A.1.

### 4.1 ZERO-SHOT TIME SERIES FORECASTING

**Setups** We first evaluate VISIONTS's **zero-shot** TSF performance without any fine-tuning on time-series modalities. To prevent data leakage and assess the out-of-distribution capabilities, we selected six widely-used datasets from the long-term TSF benchmark that are not included in MOIRAI's pre-training set for evaluation. Since most baselines cannot perform zero-shot forecasting, we report their **few-shot** results by fine-tuning on the 10% of the individual target datasets. We also evaluate the Monash benchmark (including 29 test datasets) and PF benchmark (including 6 test datasets). Notably, the Monash benchmark is more challenging for VISIONTS since they were used in MOIRAI's pre-training but not for VISIONTS. We set the hyperparameters to $r = c = 0.4$. Following common practice (Nie et al., 2022; Zhou et al., 2023; Woo et al., 2024), we conduct hyperparameter tuning on validation sets to determine the optimal context length $L$, detailed in Appendix B.1.

**Results on Long-Term TSF Benchmark** Table 1 shows that VISIONTS surprisingly achieves the best forecasting performance in most cases (7 out of 14). Specifically, VISIONTS demonstrates a relative average MSE reduction of approximately 6% compared to MOIRAI_Small and MOIRAI_Large, and performs comparably to MOIRAI_Base. When compared to the various few-shot baselines, VISIONTS shows a relative average MSE reduction ranging from 8% to 84%. Given that all baselines except for VISIONTS are trained on the time-series domain, this result is particularly encouraging. It suggests that **the transferability from images to time-series is stronger than from text to time-series, and even comparable to the in-domain transferability between time-series**. We also include a comparison with traditional algorithms (ETS, ARIMA, and Seasonal Naïve) in Appendix B.3, where VISIONTS still outperforms all of these traditional methods.

Table 1: Zero-shot or few-shot results on the long-term TSF benchmark. Results are averaged across prediction lengths {96, 192, 336, 720}, with full results in Appendix B.2. **Bold**: the best result.

| | | Zero-Shot | | | | Few-Shot (10% In-distribution Downstream Dataset) | | | | | | |
| | | Images | Time series | | | Text | | No Pretrain | | | | |
| Pretrain Method | | VISIONTS | MOIRAIS | MOIRAIB | MOIRAIL | TimeLLM | GPT4TS | DLinear | PatchTST | TimesNet | Autoformer | Informer |
|---|---|---|---|---|---|---|---|---|---|---|---|---|
| ETTh1 | MSE | **0.390** | 0.400 | 0.434 | 0.510 | 0.556 | 0.590 | 0.691 | 0.633 | 0.869 | 0.702 | 1.199 |
| | MAE | **0.414** | 0.424 | 0.439 | 0.469 | 0.522 | 0.525 | 0.600 | 0.542 | 0.628 | 0.596 | 0.809 |
| ETTh2 | MSE | **0.333** | 0.341 | 0.346 | 0.354 | 0.370 | 0.397 | 0.605 | 0.415 | 0.479 | 0.488 | 3.872 |
| | MAE | **0.375** | 0.379 | 0.382 | 0.377 | 0.394 | 0.421 | 0.538 | 0.431 | 0.465 | 0.499 | 1.513 |
| ETTm1 | MSE | **0.374** | 0.448 | 0.382 | 0.390 | 0.404 | 0.464 | 0.411 | 0.501 | 0.677 | 0.802 | 1.192 |
| | MAE | **0.372** | 0.410 | 0.388 | 0.389 | 0.427 | 0.441 | 0.429 | 0.466 | 0.537 | 0.628 | 0.821 |
| ETTm2 | MSE | 0.282 | 0.300 | **0.272** | 0.276 | 0.277 | 0.293 | 0.316 | 0.296 | 0.320 | 1.342 | 3.370 |
| | MAE | 0.321 | 0.341 | 0.321 | **0.320** | 0.323 | 0.335 | 0.368 | 0.343 | 0.353 | 0.930 | 1.440 |
| Electricity | MSE | 0.207 | 0.233 | 0.188 | 0.188 | **0.175** | 0.176 | 0.180 | 0.180 | 0.323 | 0.431 | 1.195 |
| | MAE | 0.294 | 0.320 | 0.274 | 0.273 | 0.270 | **0.269** | 0.280 | 0.273 | 0.392 | 0.478 | 0.891 |
| Weather | MSE | 0.269 | 0.242 | 0.238 | 0.260 | **0.234** | 0.238 | 0.241 | 0.242 | 0.279 | 0.300 | 0.597 |
| | MAE | 0.292 | 0.267 | **0.261** | 0.275 | 0.273 | 0.275 | 0.283 | 0.279 | 0.301 | 0.342 | 0.495 |
| **Average** | MSE | **0.309** | 0.327 | 0.310 | 0.329 | 0.336 | 0.360 | 0.407 | 0.378 | 0.491 | 0.678 | 1.904 |
| | MAE | 0.345 | 0.357 | **0.344** | 0.350 | 0.368 | 0.378 | 0.416 | 0.389 | 0.446 | 0.579 | 0.995 |
| 1st count | | 7 | 0 | 3 | 1 | 2 | 1 | 0 | 0 | 0 | 0 | 0 |

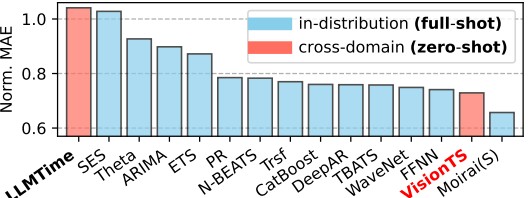

Figure 4: Aggregated results on the Monash TSF Benchmark, with full results in Appendix B.4.

| | VISIONTS | MOIRAIS | MOIRAIB | MOIRAIL |
|---|---|---|---|---|
| Electricity | **0.448** | 0.840 | 0.551 | 0.465 |
| Solar | **0.975** | 1.135 | 1.034 | 1.014 |
| Walmart | **0.225** | 0.324 | 0.291 | 0.332 |
| Weather | 0.247 | **0.229** | 0.417 | 0.331 |
| Istanbul | 0.250 | 0.294 | 0.194 | **0.186** |
| Turkey | 0.154 | 0.149 | 0.118 | **0.102** |
| 1st count | 3 | 1 | 0 | 2 |

Table 2: Results (NRMSE) on the PF benchmark, with full results in Appendix B.5.

**Results on Monash and PF Benchmarks** Fig. 4 shows the results aggregated from 29 Monash datasets, showing that VISIONTS in the zero-shot setting surpasses all models *individually* trained on each dataset (*e.g.*, FFNN, WaveNet, and TBATS) and significantly outperforms the other cross-domain baseline (*i.e.*, LLM-Time). It achieves second place among all baselines, just behind MOIRAI that pre-trained on *all* the training datasets. Table 2 shows that for the six PF datasets, where neither VISIONTS nor MOIRAI has been exposed to downstream data, VISIONTS demonstrates competitive zero-shot performance. This highlights VisionTS's strong zero-shot forecasting ability and effective cross-modality transferability.

Table 3: MAE results of TimesFM and LLMTime for zero-shot forecasting, on the last test window of the original test split.

| Method | | VISIONTS | TimesFM | LLMTime |
|---|---|---|---|---|
| ETTh1 | 96 | **0.35** | 0.45 | 0.42 |
| | 192 | **0.45** | 0.53 | 0.50 |
| ETTh2 | 96 | **0.24** | 0.35 | 0.33 |
| | 192 | **0.60** | 0.62 | 0.70 |
| ETTm1 | 96 | **0.12** | 0.19 | 0.37 |
| | 192 | **0.23** | 0.26 | 0.71 |
| ETTm2 | 96 | **0.19** | 0.24 | 0.29 |
| | 192 | **0.24** | 0.27 | 0.31 |
| **Average** | | **0.30** | 0.36 | 0.45 |

**Comparisons of TimesFM and LLMTime** Due to the relatively slow efficiency of the autoregressive decoder architecture, when compared with LLMTime (Gruver et al., 2023), Das et al. (2024) only reported results of TimesFM for the last test window on four ETT datasets. We compared VISIONTS with their results under the same setting. Table 3 shows that VISIONTS outperforms TimesFM and LLMTime in terms of MAE, indicating that image-based TSF models are on par with or even better than TS-based and text-based models.

## 4.2 FURTHER ANALYSIS OF VISIONTS

**Scaling Backbone Analysis** In Table 4 (full results in Appendix B.6), we observe that the overall performance of three MAE variants (112M, 330M, and 657M) outperforms MOIRAISmall and MOIRAILarge. Particularly, larger models show a slight decrease in performance. This may be due to **larger visual models overfitting image-specific features, reducing their transferability**. A similar

Table 4: MSE of different `MAE` variants, averaged on four prediction lengths.

| | **Base** 112M | **Large** 330M | **Huge** 657M |
|---|---|---|---|
| ETTh1 | 0.390 | **0.378** | 0.391 |
| ETTh2 | **0.333** | 0.340 | 0.339 |
| ETTm1 | **0.374** | 0.379 | 0.383 |
| ETTm2 | **0.282** | 0.286 | 0.284 |
| Electricity | 0.207 | 0.209 | **0.202** |
| Weather | **0.269** | 0.272 | 0.292 |
| Avg. | **0.309** | 0.311 | 0.315 |

Table 5: Computational cost in terms of seconds for forecasting a batch of 32 time series data.

| **Context Length** | **1k** | | | | **1k** | **2k** | **3k** | **4k** |
|---|---|---|---|---|---|---|---|---|
| **Prediction Length** | **1k** | **2k** | **3k** | **4k** | **1k** | | | |
| PatchTST | 0.01 | 0.01 | 0.01 | 0.01 | 0.01 | 0.02 | 0.03 | 0.04 |
| DeepAR | 0.26 | 0.32 | 0.37 | 0.43 | 0.26 | 4.06 | 6.10 | 8.17 |
| GPT4TS | 0.01 | 0.01 | 0.01 | 0.02 | 0.01 | 0.03 | 0.04 | 0.06 |
| MOIRAI$_{Base}$ | 0.03 | 0.04 | 0.04 | 0.05 | 0.03 | 0.04 | 0.05 | 0.06 |
| TimesFM | 0.08 | 0.14 | 0.20 | 0.27 | 0.07 | 0.13 | 0.20 | 0.25 |
| LLMTime (8B) | > 200 | | | | > 200 | | | |
| VISIONTS ($c = 0.4$) | 0.04 | 0.03 | 0.03 | 0.03 | 0.04 | 0.04 | 0.05 | 0.05 |

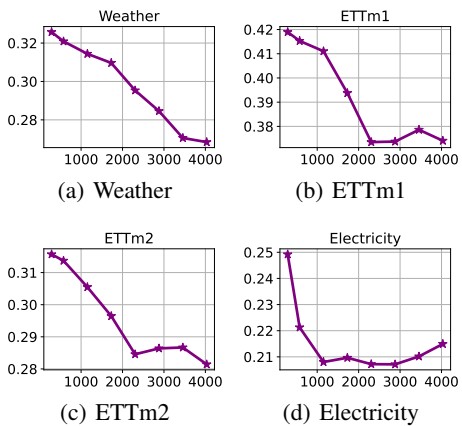

(a) Weather    (b) ETTm1

(c) ETTm2    (d) Electricity

Figure 5: MSE (Y-axis) performance of different context lengths $L$ (X-axis), averaged on four prediction lengths.

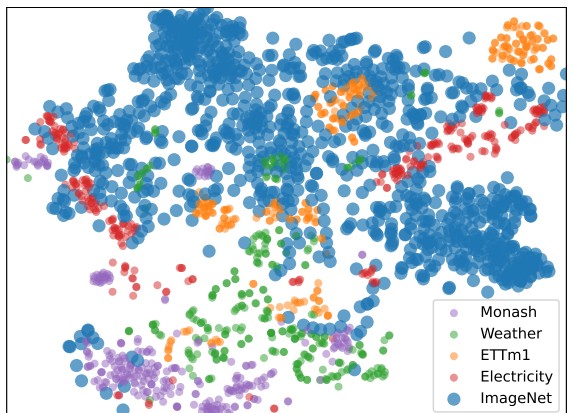

Figure 6: Modality visualization of the images (ImageNet) and time series (Monash, Weather, Electricity, and ETTm1) based on the `MAE` encoder.

phenomenon was reported in MOIRAI, where larger models were found to degrade performance. We leave the exploration of scaling laws in image-based TSF foundation models for the future. Additionally, to explore the potential with other vision models, we also test LaMa (Suvorov et al., 2022), a visual inpainting model. Results in Appendix B.6 demonstrate that VISIONTS with LaMa performs similarly to MOIRAI in the zero-shot setting. This suggests that the performance is driven by the inherent similarity between images and time series, not solely by the MAE model.

**Computational Cost** We evaluate the computation cost of different baselines on an NVIDIA A800 GPU. Results are averaged on 90 runs. Table 5 shows the results between various TSF foundation models, showing that VISIONTS are comparable to MOIRAI$_{Base}$ and GPT4TS and faster than TimesFM, which is an auto-regressive model. While computation time increases with context length for all the other Transformer-based baselines, VISIONTS remains nearly constant. This is because VISIONTS encodes input sequences into an image with constant size, ensuring $O(1)$ efficiency. In contrast, Transformer-based methods operate at $O(L^2)$ relative to context length $L$.

**Hyperparameter Analysis** Appendix B.8 illustrates the impact of three hyperparameters. For context length $L$, as shown in Fig. 5, performance typically improves with increasing $L$, particularly on high-frequency datasets like Weather (10-minute frequency) and ETTm1/ETTm2 (15-minute frequency). This aligns with other TSF foundation models like MOIRAI. As for the normalization constant $r$ and alignment constant $c$, when both of them are around 0.4, performance is generally well across most benchmarks.

**Modality Analysis: Where does the zero-shot forecastability come from?** We further examine the gap between time series and images to explain the transferability of zero-shot forecasting. We sampled 1,000 images from ImageNet-1k and 300 samples from each time series dataset. We fed

them into the `MAE`, maintaining a consistent image mask across all data. Fig. 6 visualizes the `MAE` encoder outputs of these data, which are flattened and reduced to 2-dimension by t-SNE. Notably, some time series, such as ETTm1 and Electricity, fall within the ImageNet distribution. **It suggests a relatively small gap between images and some time series, which could explain the good transferability**. Additionally, while ImageNet displays a concentrated distribution, time series are generally more scattered. For instance, ETTm1 clusters in the upper right, whereas Monash is found in the lower left, with a significant gap. **This indicates strong heterogeneity within time series data and suggests that images may serve as a bridge to connect isolated time series modality**.

**Ablation Study**  We conduct experiments to validate our choices in the Alignment step, detailed in Appendix B.7. First, we test three different interpolation strategies, which shows that **Bilinear interpolation performs best**. Second, we apply horizontal and vertical flips on the image to examine whether the assumed left-to-right, top-to-bottom order is efficient. Results show that these changes do not significantly affect performance, suggesting that **image reconstruction is isotropic and not influenced by certain orientation**.

**Qualitative Analysis: When does VISIONTS perform well, and when does it not?**  In Appendix D, we visualize the zero-shot forecasting of VISIONTS alongside the input and reconstruction images, highlighting both *successful* cases (where VISIONTS outperforms MOIRAI) and *failures* (where MOIRAI prevails). When the input exhibits strong regularity (Fig. 10), VISIONTS effectively forecasts both the periodicity (via segmentation) and trends (via `MAE`'s capabilities). In contrast, MOIRAI, akin to seasonal naïve methods, struggles to capture inter-period trends. For less-structured input (Figs. 11 to 13), MOIRAI adopts a conservative approach with lower volatility to minimize errors, while VISIONTS takes a more aggressive stance. This strategy occasionally yields more accurate trend predictions (Figs. 11 and 12) but may also result in greater MAE (Fig. 13).

### 4.3 FULL-SHOT LONG-TERM TIME SERIES FORECASTING

**Setups**  We evaluate the full-shot capability of each baseline trained on individual long-term TSF benchmarks. In addition to the six datasets used for zero-shot forecasting, we also include the popular Traffic and Illness datasets. As self-attention and feed-forward layers contain rich knowledge that can be transferred to TSF, we choose to **fine-tune only the layer normalization (LN) layers while freezing the other parameters**, which is also adopted by Zhou et al. (2023). Training details are elaborated in Appendix C.1.

**Main Results**  Table 6 summarizes the full-shot results, with standard deviations detailed in Appendix C.2. It shows that VISIONTS outperforms other baselines in most cases (46 out of 80), surpassing the non-pretrained PatchTST and the language-pretrained GPT4TS. Remarkably, except for Illness with the least data, VISIONTS demands **only a single epoch of fine-tuning**. This suggests that even minimal fine-tuning enables VisionTS to adapt to time series effectively. Compared with Table 1, fine-tuning provides limited benefits for ETTh1 and ETTh2 but significantly improves other datasets. We attribute this to the smaller data scale of ETTh1 and ETTh2.

**Ablation Study**  Tan et al. (2024) proposed several ablation variants for text-based foundation models, including **w/o LLM** (removing the LLM), **LLM2Attn/LLM2Trsf** (replacing the LLM with a single self-attention/Transformer layer), and **RandLLM** (randomly initializing the LLM). They found no significant performance differences and concluded that textual knowledge is unnecessary for TSF. We conducted similar ablations to assess the role of the vision model (VM), including **w/o VM**, **VM2Attn**, **VM2Trsf**, and **RandVM**. Table 7 with full results in Appendix C.3 shows that these variants lead to worse performance, indicating that visual knowledge is beneficial for TSF.

**Analysis: Fine-tuning strategies**  As stated before, we fine-tune only the layer normalization (LN). We also tested fine-tuning the bias, MLP, or attention layers, in addition to full fine-tuning and freezing. All hyperparameters were kept constant. Note that freezing differs from the previous zero-shot experiment, where a longer context length was used. Table 8 with full results in Appendix C.3 show that fine-tuning LN is the best. Modifying MLP or attention layers results in significant performance drops, suggesting that valuable knowledge resides in these components.

Table 6: Full-shot forecasting performance on the long-term TSF benchmark. VISIONTS is fine-tuned only a single epoch on each dataset except for Illness.

| Pretrain | Images | | Text | | | | No Pretrain | | | | | | | | | | | | | | |
|---|---|---|---|---|---|---|---|---|---|---|---|---|---|---|---|---|---|---|---|---|---|
| Method | VISIONTS | | Time-LLM | | GPT4TS | | DLinear | | PatchTST | | TimesNet | | FEDformer | | Autoformer | | Stationary | | ETSformer | | Informer | |
| Metric | MSE | MAE | MSE | MAE | MSE | MAE | MSE | MAE | MSE | MAE | MSE | MAE | MSE | MAE | MSE | MAE | MSE | MAE | MSE | MAE | MSE | MAE |
| ETTh1 96 | **0.347** | **0.376** | 0.376 | 0.402 | 0.370 | 0.389 | 0.375 | 0.399 | 0.370 | 0.399 | 0.384 | 0.402 | 0.376 | 0.419 | 0.449 | 0.459 | 0.513 | 0.491 | 0.494 | 0.479 | 0.865 | 0.713 |
| ETTh1 192 | **0.385** | **0.400** | 0.407 | 0.421 | 0.412 | 0.413 | 0.405 | 0.416 | 0.413 | 0.421 | 0.436 | 0.429 | 0.420 | 0.448 | 0.500 | 0.482 | 0.534 | 0.504 | 0.538 | 0.504 | 1.008 | 0.792 |
| ETTh1 336 | **0.407** | **0.415** | 0.430 | 0.438 | 0.448 | 0.431 | 0.439 | 0.443 | 0.422 | 0.436 | 0.491 | 0.469 | 0.459 | 0.465 | 0.521 | 0.496 | 0.588 | 0.535 | 0.574 | 0.521 | 1.107 | 0.809 |
| ETTh1 720 | **0.439** | **0.443** | 0.457 | 0.468 | 0.441 | 0.449 | 0.472 | 0.490 | 0.447 | 0.466 | 0.521 | 0.500 | 0.506 | 0.507 | 0.514 | 0.512 | 0.643 | 0.616 | 0.562 | 0.535 | 1.181 | 0.865 |
| ETTh1 avg | **0.395** | **0.409** | 0.418 | 0.432 | 0.418 | 0.421 | 0.423 | 0.437 | 0.413 | 0.431 | 0.458 | 0.450 | 0.440 | 0.460 | 0.496 | 0.487 | 0.570 | 0.537 | 0.542 | 0.510 | 1.040 | 0.795 |
| ETTh2 96 | **0.269** | **0.328** | 0.286 | 0.346 | 0.280 | 0.335 | 0.289 | 0.353 | 0.274 | 0.336 | 0.340 | 0.374 | 0.358 | 0.397 | 0.346 | 0.388 | 0.476 | 0.458 | 0.340 | 0.391 | 3.755 | 1.525 |
| ETTh2 192 | **0.332** | **0.374** | 0.361 | 0.391 | 0.348 | 0.380 | 0.383 | 0.418 | 0.339 | 0.379 | 0.402 | 0.414 | 0.429 | 0.439 | 0.456 | 0.452 | 0.512 | 0.493 | 0.430 | 0.439 | 5.602 | 1.931 |
| ETTh2 336 | 0.351 | 0.395 | 0.390 | 0.414 | 0.380 | 0.405 | 0.448 | 0.465 | **0.329** | 0.380 | 0.452 | 0.452 | 0.496 | 0.487 | 0.482 | 0.486 | 0.552 | 0.551 | 0.485 | 0.479 | 4.721 | 1.835 |
| ETTh2 720 | 0.390 | 0.430 | 0.405 | 0.434 | 0.406 | 0.436 | 0.605 | 0.551 | **0.379** | **0.422** | 0.462 | 0.468 | 0.463 | 0.474 | 0.515 | 0.511 | 0.562 | 0.560 | 0.500 | 0.497 | 3.647 | 1.625 |
| ETTh2 avg | 0.336 | 0.382 | 0.361 | 0.396 | 0.354 | 0.389 | 0.431 | 0.447 | **0.330** | **0.379** | 0.414 | 0.427 | 0.437 | 0.449 | 0.450 | 0.459 | 0.526 | 0.516 | 0.439 | 0.452 | 4.431 | 1.729 |
| ETTm1 96 | **0.281** | **0.322** | 0.291 | 0.341 | 0.300 | 0.340 | 0.299 | 0.343 | 0.290 | 0.342 | 0.338 | 0.375 | 0.379 | 0.419 | 0.505 | 0.475 | 0.386 | 0.398 | 0.375 | 0.398 | 0.672 | 0.571 |
| ETTm1 192 | **0.322** | **0.353** | 0.341 | 0.369 | 0.343 | 0.368 | 0.335 | 0.365 | 0.332 | 0.369 | 0.374 | 0.387 | 0.426 | 0.441 | 0.553 | 0.496 | 0.459 | 0.444 | 0.408 | 0.410 | 0.795 | 0.669 |
| ETTm1 336 | **0.356** | **0.379** | 0.359 | 0.379 | 0.376 | 0.386 | 0.369 | 0.386 | 0.366 | 0.392 | 0.410 | 0.411 | 0.445 | 0.459 | 0.621 | 0.537 | 0.495 | 0.464 | 0.435 | 0.428 | 1.212 | 0.871 |
| ETTm1 720 | **0.391** | **0.413** | 0.433 | 0.419 | 0.431 | 0.416 | 0.425 | 0.421 | 0.416 | 0.420 | 0.478 | 0.450 | 0.543 | 0.490 | 0.671 | 0.561 | 0.585 | 0.516 | 0.499 | 0.462 | 1.166 | 0.823 |
| ETTm1 avg | **0.338** | **0.367** | 0.356 | 0.377 | 0.363 | 0.378 | 0.357 | 0.379 | 0.351 | 0.381 | 0.400 | 0.406 | 0.448 | 0.452 | 0.588 | 0.517 | 0.481 | 0.456 | 0.429 | 0.425 | 0.961 | 0.734 |
| ETTm2 96 | 0.169 | 0.256 | **0.162** | **0.248** | 0.163 | 0.249 | 0.167 | 0.269 | 0.165 | 0.255 | 0.187 | 0.267 | 0.203 | 0.287 | 0.255 | 0.339 | 0.192 | 0.274 | 0.189 | 0.280 | 0.365 | 0.453 |
| ETTm2 192 | 0.225 | 0.294 | 0.235 | 0.304 | 0.222 | **0.291** | 0.224 | 0.303 | **0.220** | 0.292 | 0.249 | 0.309 | 0.269 | 0.328 | 0.281 | 0.340 | 0.280 | 0.339 | 0.253 | 0.319 | 0.533 | 0.563 |
| ETTm2 336 | 0.278 | 0.334 | 0.280 | 0.329 | **0.273** | **0.327** | 0.281 | 0.342 | 0.274 | 0.329 | 0.321 | 0.351 | 0.325 | 0.366 | 0.339 | 0.372 | 0.334 | 0.361 | 0.314 | 0.357 | 1.363 | 0.887 |
| ETTm2 720 | 0.372 | 0.392 | 0.366 | 0.382 | **0.357** | **0.376** | 0.397 | 0.421 | 0.362 | 0.385 | 0.408 | 0.403 | 0.421 | 0.415 | 0.433 | 0.432 | 0.417 | 0.413 | 0.414 | 0.413 | 3.379 | 1.338 |
| ETTm2 avg | 0.261 | 0.319 | 0.261 | 0.316 | **0.254** | **0.311** | 0.267 | 0.334 | 0.255 | 0.315 | 0.291 | 0.333 | 0.305 | 0.349 | 0.327 | 0.371 | 0.306 | 0.347 | 0.293 | 0.342 | 1.410 | 0.810 |
| Illness 24 | 2.034 | 0.937 | 1.792 | 0.807 | 1.869 | 0.823 | 2.215 | 1.081 | **1.319** | **0.754** | 2.317 | 0.934 | 3.228 | 1.260 | 3.483 | 1.287 | 2.294 | 0.945 | 2.527 | 1.020 | 5.764 | 1.677 |
| Illness 36 | 1.866 | 0.888 | 1.833 | **0.833** | 1.853 | 0.854 | 1.963 | 0.963 | **1.430** | 0.834 | 1.972 | 0.920 | 2.679 | 1.080 | 3.103 | 1.148 | 1.825 | 0.848 | 2.615 | 1.007 | 4.755 | 1.467 |
| Illness 48 | 1.784 | 0.870 | 2.269 | 1.012 | 1.886 | 0.855 | 2.130 | 1.024 | **1.553** | **0.815** | 2.238 | 0.940 | 2.622 | 1.078 | 2.669 | 1.085 | 2.010 | 0.900 | 2.359 | 0.972 | 4.763 | 1.469 |
| Illness 60 | 1.910 | 0.912 | 2.177 | 0.925 | 1.877 | 0.877 | 2.368 | 1.096 | **1.470** | **0.788** | 2.027 | 0.928 | 2.857 | 1.157 | 2.770 | 1.125 | 2.178 | 0.963 | 2.487 | 1.016 | 5.264 | 1.564 |
| Illness avg | 1.899 | 0.902 | 2.018 | 0.894 | 1.871 | 0.852 | 2.169 | 1.041 | **1.443** | **0.798** | 2.139 | 0.931 | 2.847 | 1.144 | 3.006 | 1.161 | 2.077 | 0.914 | 2.497 | 1.004 | 5.137 | 1.544 |
| Weather 96 | **0.142** | 0.192 | 0.155 | 0.199 | 0.148 | **0.188** | 0.176 | 0.237 | 0.149 | 0.198 | 0.172 | 0.220 | 0.217 | 0.296 | 0.266 | 0.336 | 0.173 | 0.223 | 0.197 | 0.281 | 0.300 | 0.384 |
| Weather 192 | **0.191** | 0.238 | 0.223 | 0.261 | 0.192 | **0.230** | 0.220 | 0.282 | 0.194 | 0.241 | 0.219 | 0.261 | 0.276 | 0.336 | 0.307 | 0.367 | 0.245 | 0.285 | 0.237 | 0.312 | 0.598 | 0.544 |
| Weather 336 | 0.246 | 0.282 | 0.251 | 0.279 | 0.246 | **0.273** | 0.265 | 0.319 | **0.245** | 0.282 | 0.280 | 0.306 | 0.339 | 0.380 | 0.359 | 0.395 | 0.321 | 0.338 | 0.298 | 0.353 | 0.578 | 0.523 |
| Weather 720 | 0.328 | 0.337 | 0.345 | 0.342 | 0.320 | **0.328** | 0.333 | 0.362 | **0.314** | 0.334 | 0.365 | 0.359 | 0.403 | 0.428 | 0.419 | 0.428 | 0.414 | 0.410 | 0.352 | 0.388 | 1.059 | 0.741 |
| Weather avg | 0.227 | 0.262 | 0.244 | 0.270 | 0.227 | **0.255** | 0.249 | 0.300 | **0.226** | 0.264 | 0.259 | 0.287 | 0.309 | 0.360 | 0.338 | 0.382 | 0.288 | 0.314 | 0.271 | 0.334 | 0.634 | 0.548 |
| Traffic 96 | **0.344** | **0.236** | 0.392 | 0.267 | 0.396 | 0.264 | 0.410 | 0.282 | 0.360 | 0.249 | 0.593 | 0.321 | 0.587 | 0.366 | 0.613 | 0.388 | 0.612 | 0.338 | 0.607 | 0.392 | 0.719 | 0.391 |
| Traffic 192 | **0.372** | **0.249** | 0.409 | 0.271 | 0.412 | 0.268 | 0.423 | 0.287 | 0.379 | 0.256 | 0.617 | 0.336 | 0.604 | 0.373 | 0.616 | 0.382 | 0.613 | 0.340 | 0.621 | 0.399 | 0.696 | 0.379 |
| Traffic 336 | **0.383** | **0.257** | 0.434 | 0.296 | 0.421 | 0.273 | 0.436 | 0.296 | 0.392 | 0.264 | 0.629 | 0.336 | 0.621 | 0.383 | 0.622 | 0.337 | 0.618 | 0.328 | 0.622 | 0.396 | 0.777 | 0.420 |
| Traffic 720 | **0.422** | **0.280** | 0.451 | 0.291 | 0.455 | 0.291 | 0.466 | 0.315 | 0.432 | 0.286 | 0.640 | 0.350 | 0.626 | 0.382 | 0.660 | 0.408 | 0.653 | 0.355 | 0.632 | 0.396 | 0.864 | 0.472 |
| Traffic avg | **0.380** | **0.256** | 0.422 | 0.281 | 0.421 | 0.274 | 0.434 | 0.295 | 0.391 | 0.264 | 0.620 | 0.336 | 0.610 | 0.376 | 0.628 | 0.379 | 0.624 | 0.340 | 0.621 | 0.396 | 0.764 | 0.416 |
| Electricity 96 | **0.126** | **0.218** | 0.137 | 0.233 | 0.141 | 0.239 | 0.140 | 0.237 | 0.129 | 0.222 | 0.168 | 0.272 | 0.193 | 0.308 | 0.201 | 0.317 | 0.169 | 0.273 | 0.187 | 0.304 | 0.274 | 0.368 |
| Electricity 192 | **0.144** | **0.237** | 0.152 | 0.247 | 0.158 | 0.253 | 0.153 | 0.249 | 0.157 | 0.240 | 0.184 | 0.289 | 0.201 | 0.315 | 0.222 | 0.334 | 0.182 | 0.286 | 0.199 | 0.315 | 0.296 | 0.386 |
| Electricity 336 | **0.162** | **0.256** | 0.169 | 0.267 | 0.172 | 0.266 | 0.169 | 0.267 | 0.163 | 0.259 | 0.198 | 0.300 | 0.214 | 0.329 | 0.231 | 0.338 | 0.200 | 0.304 | 0.212 | 0.329 | 0.300 | 0.394 |
| Electricity 720 | **0.192** | **0.286** | 0.200 | 0.290 | 0.207 | 0.293 | 0.203 | 0.301 | 0.197 | 0.290 | 0.220 | 0.320 | 0.246 | 0.355 | 0.254 | 0.361 | 0.222 | 0.321 | 0.233 | 0.345 | 0.373 | 0.439 |
| Electricity avg | **0.156** | **0.249** | 0.165 | 0.259 | 0.170 | 0.263 | 0.166 | 0.264 | 0.162 | 0.253 | 0.193 | 0.295 | 0.214 | 0.327 | 0.227 | 0.338 | 0.193 | 0.296 | 0.208 | 0.323 | 0.311 | 0.397 |
| 1st count | 46 | | 4 | | 12 | | 0 | | 19 | | 0 | | 0 | | 0 | | 0 | | 0 | | 0 | |

Table 7: MSE results for ablation studies, averaged on four prediction lengths.

| - | w/o VM | VM2Attn | VM2Trsf | Rand-VM |
|---|---|---|---|---|
| ETTh1 | **0.395** | 0.785 | 0.448 | 0.459 | 0.534 |
| ETTh2 | **0.336** | 0.420 | 0.418 | 0.448 | 0.411 |
| ETTm1 | **0.338** | 0.676 | 0.397 | 0.398 | 0.433 |
| ETTm2 | **0.261** | 0.379 | 0.274 | 0.292 | 0.288 |
| Avg. | **0.333** | 0.565 | 0.384 | 0.399 | 0.417 |

Table 8: MSE for different fine-tuning strategies, averaged on four prediction lengths.

| | All | LN | Bias | MLP | Attn | Freeze |
|---|---|---|---|---|---|---|
| ETTh1 | 0.534 | **0.395** | 0.401 | 0.534 | 0.554 | 0.419 |
| ETTh2 | 0.411 | **0.336** | 0.347 | 0.401 | 0.392 | 0.340 |
| ETTm1 | 0.433 | **0.338** | 0.343 | 0.441 | 0.444 | 0.374 |
| ETTm2 | 0.288 | 0.261 | **0.256** | 0.292 | 0.289 | 0.305 |
| Avg. | 0.417 | **0.333** | 0.337 | 0.417 | 0.420 | 0.360 |

## 5 RELATED WORK

Depending on the pre-training data, TSF foundation models can be categorized into Text-based and TS-based. We first review these related works, and then introduce recent research for image-based time series analysis.

**Text-based TSF Foundation Models**  Large Language Models (LLMs) pre-trained on large amounts of text data are being applied to TSF tasks. For example, Zhou et al. (2023) fine-tuned a pre-trained GPT (Radford et al., 2019) on each time-series downstream task, such as forecasting, classification, imputation, and anomaly detection. Based on Llama (Touvron et al., 2023), Jin et al. (2024) froze the pre-trained LLM and reprogrammed the time series to align with the language modality. Bian et al. (2024) adopted a two-stage approach by continually pre-training GPT (Radford et al., 2019) on the time-series domain. Nevertheless, the TSF performance of LLMs has recently been questioned by Tan et al. (2024), which designed several ablation studies to show that textual knowledge is unnecessary for forecasting. In this paper, we attribute it to the large modality gap. Some recent approaches focus on directly transforming the time series into natural texts for LLMs, allowing for zero-shot forecasting. For example, PromptCast (Xue & Salim, 2023) used pre-defined

templates to describe numerical time series data, while LLMTime (Gruver et al., 2023) directly separated time steps using commas and separates digits using spaces to construct the text input. However, due to the efficiency issue of the autoregressive decoding strategy and the expensive inference cost of large language models, their practical use is limited.

**Time Series-Based TSF Foundation Models**    Self-supervised pre-training a TSF model on the same dataset used for downstream TSF tasks is a well-explored topic (Ma et al., 2023; Zhang et al., 2024), such as denoising autoencoders (Zerveas et al., 2021) or contrastive learning (Woo et al., 2022a; Yue et al., 2022). They follow a similar paradigm to the masked autoencoder (`MAE`) in computer vision. However, these methods rarely examine the cross-dataset generalization capabilities. Recently, research has shifted towards training universal foundation models, by collecting large-scale time series datasets from diverse domains (Goswami et al., 2024; Liu et al., 2024; Das et al., 2024; Dong et al., 2024; Feng et al., 2024) or generating numerous synthetic time series data (Fu et al., 2024; Yang et al., 2024). As a representative method, Woo et al. (2024) collected 27 billion observations across nine domains and trained TSF foundation models of various scales, achieving strong zero-shot performance. However, given the severe heterogeneity, constructing high-quality large datasets poses significant challenges for building these foundation models.

**Image-Based Time-Series Analysis**    Previous research has investigated encoding time series data into images and used convolutional neural networks (CNNs) trained from scratch for classification (Wang & Oates, 2015a;b; Hatami et al., 2018) or forecasting (Li et al., 2020; Sood et al., 2021; Semenoglou et al., 2023). Recent researchers explored using pre-trained models for these imaging time series. Li et al. (2024) used a pre-trained vision transformer (ViT) for classification. Wimmer & Rekabsaz (2023) and Zhang et al. (2023) employed vision-language multimodal pre-trained models to extract predictive features and generate text descriptions. Yang et al. (2024) generated synthetic time series data to pre-train a vision model for the TSF task. However, these studies did not deeply examine the transferability from natural images to TSF. Despite early efforts by Zhou et al. (2023) to fine-tune a BEiT (Bao et al., 2022) trained on images for time series forecasting, it still falls short of the leading text-based and TS-based TSF foundation models. To the best of our knowledge, we are the first to show that an image-based foundation model, without further time-series adaptation, can match or even surpass other types of TSF foundation models.

# 6    CONCLUSION

In this paper, we explore a novel approach to building a time series forecasting (TSF) foundation model using natural images, offering a new perspective distinct from the traditional text-based and TS-based methods. By leveraging the intrinsic similarities between images and time series, we introduced VISIONTS, an `MAE`-based TSF foundation model that reformulates the TSF task as an image reconstruction problem. Our extensive evaluations demonstrate that VISIONTS achieves outstanding forecasting performance in zero-shot and full-shot settings, being a free lunch for a TSF foundation model. We hope our findings could open new avenues for further cross-modality research.

# 7    LIMITATION AND FUTURE WORK

- **Exploring Other Architectures**: As a preliminary study, we employed a basic `MAE` model. Utilizing more advanced models like diffusion models (Rombach et al., 2022; Peebles & Xie, 2023) presents a promising research direction.

- **Expanding Time Series Capacities**: Due to limitations in the visual model, VISIONTS cannot utilize exogenous covariates and perform distribution forecasting. Future modifications to the model structure may empower it with more time series capabilities.

- **Continual Pretraining**: As discussed in Table 4, larger visual models may overfit image-specific features, limiting their transferability to time series. Investigating whether continual pretraining on large-scale time series can reduce the gap between the two modalities is an interesting avenue.

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

# A    DETAILS OF EXPERIMENTS

## A.1    BENCHMARK AND BASELINES

**Long-Term TSF Benchmark**    We evaluate our model on 8 widely used long-term TSF datasets (Zhou et al., 2021; Wu et al., 2021), including ETTh1, ETTh2, ETTm1, ETTm2, Electricity, Traffic, Illness, and Weather. Performance is assessed using Mean Squared Error (MSE) and Mean Absolute Error (MAE), with lower values indicating better forecasting accuracy.

**Monash Benchmark**    Following Woo et al. (2024), we tested 29 Monash datasets (Godahewa et al., 2021) using GluonTS (Alexandrov et al., 2020), including M1 Monthly, M3 Monthly, M3 Other, M4 Monthly, M4 Weekly, M4 Daily, M4 Hourly, Tourism Quarterly, Tourism Monthly, CIF 2016, Australian Electricity Demand, Bitcoin, Pedestrian Counts, Vehicle Trips, KDD Cup, Weather, NN5 Daily, NN5 Weekly, Carparts, FRED-MD, Traffic Hourly, Traffic Weekly, Rideshare, Hospital, COVID Deaths, Temperature Rain, Sunspot, Saugeen River Flow, and US Births. Performance is assessed using MAE.

**PF Benchmark**    Woo et al. (2024) tested their methods on six datasets for evaluating the probability forecasting ability (PF), including Electricity, Solar, Walmart, Weather, Istanbul Traffic, and Turkey Power. Since MAE cannot output distributions, we report the point forecasting metrics on these six PF datasets, including the symmetric mean absolute percentage error (sMAPE), mean absolute scaled error (MASE) (Hyndman & Koehler, 2006), normalized deviation (ND), and normalized root mean squared error (NRMSE) (Yu et al., 2016).

**Baselines**    The baseline models selected for comparison are briefly described below:

1. **MOIRAI** (Woo et al., 2024) is a TSF foundation model trained on the Large-scale Open Time Series Archive (LOTSA), with over 27B observations across nine domains. It has three variants: **small**, **base**, and **large**.
2. **TimesFM** (Das et al., 2024) is a decoder-style TSF foundation model, using a large time-series corpus comprising both real-world and synthetic datasets.
3. **Time-LLM** (Jin et al., 2024) is a text-based TSF foundation model built on Llama, which reprograms time series data to align with the language modality, keeping the LLM frozen.
4. **GPT4TS** (Zhou et al., 2023) (OneFitsAll) is another text-based model based on GPT, fine-tuned for forecasting tasks.
5. **LLMTime** (Gruver et al., 2023) encodes time series data to a text sequence, supporting zero-shot forecasting.
6. **DLinear** (Zeng et al., 2023) proposes a linear forecasting model, enhanced by seasonal-trend decomposition or normalization.
7. **PatchTST** (Nie et al., 2022) uses Transformer encoders with patching and channel independence techniques for improved predictions.
8. **TimesNet** (Wu et al., 2023) applies convolution kernels along the time dimension, using temporal decomposition and periodical segmentation to capture temporal patterns.
9. **FEDformer** (Zhou et al., 2022) employs a sparse frequency domain representation, using frequency-enhanced blocks for cross-time dependency.
10. **Autoformer** (Wu et al., 2021) uses series decomposition blocks and Auto-Correlation to capture cross-time dependency.
11. **Stationary** (Liu et al., 2022) introduces stationarization and de-stationary attention mechanisms.
12. **ETSFormer** (Woo et al., 2022b) leverages exponential smoothing principles, including exponential smoothing and frequency attention mechanisms.
13. **Informer** (Zhou et al., 2021) proposes ProbSparse self-attention and distillation operations.

For the long-term TSF benchmark, we include TS-based foundation model results from their original papers, Text-based model results from Tan et al. (2024), and other baseline results from Zhou et al. (2023). For the Monash and PF benchmark, we include results from Woo et al. (2024).

Table 9: Periodicity ($P$) search range for the sampling frequency. $x$ denotes the number of sampling frequencies. For example, for data with a sampling frequency of 2 minutes (2T), we have $x = 2$, and the possible search range of $P$ is $\{1440/x, 10080/x, 1\} = \{720, 5040, 1\}$.

| Sampling Frequency | Possible Seasonalities | Possible P |
|---|---|---|
| Second (S) | 1 hour | $\{3600/x, 1\}$ |
| Minute (T) | 1 day or 1 week | $\{1440/x, 10080/x, 1\}$ |
| Hour (H) | 1 day or 1 week | $\{24/x, 168/x, 1\}$ |
| Day (D) | 1 week, 1 month, or 1 year | $\{7/x, 30/x, 365/x, 1\}$ |
| Week (W) | 1 year or 1 month | $\{52/x, 4/x, 1\}$ |
| Month (M) | 1 year, 6 months, or 3 months | $\{12/x, 6/x, 3/x, 1\}$ |
| Business Day (B) | 1 week | $\{5/x, 1\}$ |
| Quarter (Q) | 1 year or 6 months | $\{4/x, 2/x, 1\}$ |
| Others | - | $\{1\}$ |

Table 10: Final $P$ used for each dataset in our experiment.

| | Frequency | P | Datasets | | | |
|---|---|---|---|---|---|---|
| | H | 24 | ETTh1 | ETTh2 | Electricity | Traffic |
| | W | 52 | Illness | | | |
| Long-Term TSF | 15T | 96 | ETTm1 | ETTm2 | | |
| | 10T | 144 | Weather | | | |
| | H | 24 | Electricity | Solar | Istanbul Traffic | Turkey Power |
| PF | W | 52 | Walmart | | | |
| | 10T | 144 | Weather | | | |
| | D | 1 | M4 Daily | COVID Deaths | | |
| | W | 1 | NN5 Weekly | | | |
| | M | 1 | FRED-MD | | | |
| | Q | 1 | M3 Other | | | |
| | M | 3 | M3 Monthly | M4 Monthly | CIF 2016 (6) | |
| | W | 4 | M4 Weekly | Traffic Weekly | | |
| | Q | 4 | Tourism Quarterly | | | |
| Monash | M | 6 | CIF 2016 (12) | Car Parts | | |
| | D | 7 | Bitcoin | Vehicle Trips | Weather | NN5 Daily |
| | D | 7 | US Births | Saugeen Day | Temperature Rain | |
| | M | 12 | Tourism Monthly | Hospital | M1 Monthly | |
| | H | 24 | M4 Hourly | KDD cup | Pedestrian Counts | |
| | H | 24 | Traffic Hourly | Rideshare | | |
| | D | 30 | Sunspot | | | |
| | 0.5H | 336 | Aus. Elec. Demand | | | |

Table 11: Comparison of setting $P = 1$ for VISIONTS.

| | VISIONTS | | $P = 1$ | |
|---|---|---|---|---|
| | MSE | MAE | MSE | MAE |
| ETTh1 | 0.390 | 0.414 | 0.840 | 0.628 |
| ETTh2 | 0.333 | 0.375 | 0.424 | 0.445 |
| ETTm1 | 0.374 | 0.372 | 0.660 | 0.533 |
| ETTm2 | 0.282 | 0.321 | 0.312 | 0.363 |
| **Average** | 0.344 | 0.370 | 0.559 | 0.492 |

**Environment** All experiments are conducted using *Time-Series-Library* (https://github.com/thuml/Time-Series-Library) and GluonTS library (Alexandrov et al., 2020) on an NVIDIA A800 GPU.

## A.2 PERIODICITY SELECTION

We first determine a range of period lengths based on the sampling frequency of the data, shown in Table 9. This frequency-based strategy is also employed by Alexandrov et al. (2020) while we extend the search range for tuning. We select the optimal $P$ from this range on the validation set. The final $P$ used in our experiments are summarized in Table 10.

To demonstrate the influence of $P$ and the effectiveness of our periodicity selection strategy, we set $P = 1$ and compare the results with the above strategy. Table 11 shows that such strategy (denoted as VISIONTS) significantly outperforms the naive strategy that sets $P = 1$.

# B   ZERO-SHOT FORECASTING

## B.1   HYPERPARAMETERS

Table 12: Hyperparameters for VISIONTS used in our zero-shot forecasting (Long-term TSF).

|  | ETTh1 | ETTh2 | ETTm1 | ETTm2 | Weather | Electricity |
|---|---|---|---|---|---|---|
| Normalization constant $r$ | 0.4 | 0.4 | 0.4 | 0.4 | 0.4 | 0.4 |
| Alignment constant $c$ | 0.4 | 0.4 | 0.4 | 0.4 | 0.4 | 0.4 |
| Context length $L$ | 2880 | 1728 | 2304 | 4032 | 4032 | 2880 |

We conduct hyperparameter tuning on validation sets to determine the optimal context length $L$. Final used hyperparameters are summarized in Table 12.

## B.2   FULL FORECASTING RESULTS OF THE LONG-TERM TSF BENCHMARK

Table 13: Full results of Table 1: Zero-shot or few-shot results on the long-term TSF benchmark. **Bold**: the best result.

| | | Zero-Shot | | | Few-Shot (10% Downstream Dataset) | | | | | | |
| | | Images | Time-series | | | Text | | No Pretrain | | | | |
| Pretrain Method | | VISIONTIME | MOIRAI$_S$ | MOIRAI$_B$ | MOIRAI$_L$ | TimeLLM | GPT4TS | DLinear | PatchTST | TimesNet | Autoformer | Informer |
| Metric | | MSE MAE | MSE MAE | MSE MAE | MSE MAE | MSE MAE | MSE MAE | MSE MAE | MSE MAE | MSE MAE | MSE MAE | MSE MAE |
| ETTh1 | 96 | **0.353 0.383** | 0.375 0.402 | 0.384 0.402 | 0.380 0.398 | 0.448 0.460 | 0.458 0.456 | 0.492 0.495 | 0.516 0.485 | 0.861 0.628 | 0.613 0.552 | 1.179 0.792 |
| | 192 | **0.392 0.410** | 0.399 0.419 | 0.425 0.429 | 0.440 0.434 | 0.484 0.483 | 0.570 0.516 | 0.565 0.538 | 0.598 0.524 | 0.797 0.593 | 0.722 0.598 | 1.199 0.806 |
| | 336 | **0.407 0.423** | 0.412 0.429 | 0.456 0.450 | 0.514 0.474 | 0.589 0.540 | 0.608 0.535 | 0.721 0.622 | 0.657 0.550 | 0.941 0.648 | 0.750 0.619 | 1.202 0.811 |
| | 720 | **0.406 0.441** | 0.413 0.444 | 0.470 0.473 | 0.705 0.568 | 0.700 0.604 | 0.725 0.591 | 0.986 0.743 | 0.762 0.610 | 0.877 0.641 | 0.721 0.616 | 1.217 0.825 |
| | avg | **0.390 0.414** | 0.400 0.424 | 0.434 0.439 | 0.510 0.469 | 0.556 0.522 | 0.590 0.525 | 0.691 0.600 | 0.633 0.542 | 0.869 0.628 | 0.702 0.596 | 1.199 0.809 |
| ETTh2 | 96 | **0.271** 0.328 | 0.281 0.334 | 0.277 0.327 | 0.287 **0.325** | 0.275 0.326 | 0.331 0.374 | 0.357 0.411 | 0.353 0.389 | 0.378 0.409 | 0.413 0.451 | 3.837 1.508 |
| | 192 | **0.328 0.367** | 0.340 0.373 | 0.340 0.374 | 0.347 0.367 | 0.374 0.373 | 0.402 0.411 | 0.569 0.519 | 0.403 0.414 | 0.490 0.467 | 0.474 0.477 | 3.856 1.513 |
| | 336 | **0.345 0.381** | 0.362 0.393 | 0.371 0.401 | 0.377 0.393 | 0.406 0.429 | 0.406 0.433 | 0.671 0.572 | 0.426 0.441 | 0.537 0.494 | 0.547 0.543 | 3.952 1.526 |
| | 720 | **0.388** 0.422 | 0.380 0.416 | 0.394 0.424 | 0.404 **0.421** | 0.427 0.449 | 0.449 0.464 | 0.824 0.648 | 0.477 0.480 | 0.510 0.491 | 0.516 0.523 | 3.842 1.503 |
| | avg | **0.333 0.375** | 0.341 0.379 | 0.346 0.382 | 0.354 0.377 | 0.370 0.394 | 0.397 0.421 | 0.605 0.538 | 0.415 0.431 | 0.479 0.465 | 0.488 0.499 | 3.872 1.513 |
| ETTm1 | 96 | **0.341 0.347** | 0.404 0.383 | 0.335 0.360 | 0.353 0.363 | 0.346 0.388 | 0.390 0.404 | 0.352 0.392 | 0.410 0.419 | 0.583 0.501 | 0.774 0.614 | 1.162 0.785 |
| | 192 | **0.360 0.360** | 0.435 0.402 | 0.366 0.379 | 0.376 0.380 | 0.373 0.416 | 0.429 0.423 | 0.382 0.412 | 0.437 0.434 | 0.630 0.528 | 0.754 0.592 | 1.172 0.793 |
| | 336 | **0.377 0.374** | 0.462 0.416 | 0.391 0.394 | 0.399 0.395 | 0.413 0.426 | 0.469 0.439 | 0.419 0.434 | 0.476 0.454 | 0.725 0.568 | 0.869 0.677 | 1.227 0.908 |
| | 720 | **0.416 0.405** | 0.490 0.437 | 0.434 0.419 | 0.432 0.417 | 0.485 0.476 | 0.569 0.498 | 0.490 0.477 | 0.681 0.556 | 0.769 0.549 | 0.810 0.630 | 1.207 0.797 |
| | avg | **0.374 0.372** | 0.448 0.410 | 0.382 0.388 | 0.390 0.389 | 0.404 0.427 | 0.464 0.441 | 0.411 0.429 | 0.501 0.466 | 0.677 0.537 | 0.802 0.628 | 1.192 0.821 |
| ETTm2 | 96 | 0.228 0.282 | 0.205 0.282 | 0.195 0.269 | **0.189 0.260** | 0.177 0.261 | 0.188 0.269 | 0.213 0.303 | 0.191 0.274 | 0.212 0.285 | 0.352 0.454 | 3.203 1.407 |
| | 192 | 0.262 0.305 | 0.261 0.318 | 0.247 0.303 | 0.247 **0.300** | 0.241 0.314 | 0.251 0.309 | 0.278 0.345 | 0.252 0.317 | 0.270 0.323 | 0.694 0.691 | 3.112 1.387 |
| | 336 | 0.293 **0.328** | 0.319 0.355 | **0.291** 0.333 | 0.295 0.334 | 0.274 0.327 | 0.307 0.346 | 0.338 0.385 | 0.306 0.353 | 0.323 0.353 | 2.408 1.407 | 3.255 1.421 |
| | 720 | **0.343 0.370** | 0.415 0.410 | 0.355 0.377 | 0.372 0.386 | 0.417 0.390 | 0.426 0.417 | 0.436 0.440 | 0.433 0.427 | 0.474 0.449 | 1.913 1.166 | 3.909 1.543 |
| | avg | 0.282 0.321 | 0.300 0.341 | **0.272** 0.321 | 0.276 **0.320** | 0.277 0.323 | 0.293 0.335 | 0.316 0.368 | 0.296 0.343 | 0.320 0.353 | 1.342 0.930 | 3.370 1.440 |
| Electricity | 96 | 0.177 0.266 | 0.205 0.299 | 0.158 0.248 | 0.152 0.242 | **0.139** 0.241 | **0.139 0.237** | 0.150 0.253 | 0.140 0.238 | 0.299 0.373 | 0.261 0.348 | 1.259 0.919 |
| | 192 | 0.188 0.277 | 0.220 0.310 | 0.174 0.263 | 0.171 0.259 | **0.151 0.248** | 0.156 0.252 | 0.164 0.264 | 0.160 0.255 | 0.305 0.379 | 0.338 0.406 | 1.160 0.873 |
| | 336 | 0.207 0.296 | 0.236 0.323 | 0.191 0.278 | 0.192 0.278 | **0.169 0.270** | 0.175 **0.270** | 0.181 0.282 | 0.180 0.276 | 0.319 0.391 | 0.410 0.474 | 1.157 0.872 |
| | 720 | 0.256 0.337 | 0.270 0.347 | 0.229 0.307 | 0.236 0.313 | 0.240 0.322 | **0.233** 0.317 | 0.223 0.321 | 0.241 0.323 | 0.369 0.426 | 0.715 0.685 | 1.203 0.898 |
| | avg | 0.207 0.294 | 0.233 0.320 | 0.188 0.274 | 0.188 0.273 | **0.175** 0.270 | 0.176 **0.269** | 0.180 0.280 | 0.180 0.273 | 0.323 0.392 | 0.431 0.478 | 1.195 0.891 |
| Weather | 96 | 0.220 0.257 | 0.173 0.212 | 0.167 **0.203** | 0.177 0.208 | **0.161** 0.210 | 0.163 0.215 | 0.171 0.224 | 0.165 0.215 | 0.184 0.230 | 0.221 0.297 | 0.374 0.401 |
| | 192 | 0.244 0.275 | 0.216 0.250 | 0.209 **0.241** | 0.219 0.249 | **0.204** 0.248 | 0.210 0.254 | 0.215 0.263 | 0.210 0.257 | 0.245 0.283 | 0.270 0.322 | 0.552 0.478 |
| | 336 | 0.280 0.299 | 0.260 0.282 | **0.256 0.276** | 0.277 0.292 | 0.261 0.302 | **0.256** 0.292 | 0.258 0.299 | 0.259 0.297 | 0.305 0.321 | 0.320 0.351 | 0.724 0.541 |
| | 720 | 0.330 0.337 | 0.320 0.322 | 0.321 0.323 | 0.365 0.350 | **0.309** 0.332 | 0.321 0.339 | 0.320 0.346 | 0.332 0.346 | 0.381 0.371 | 0.390 0.396 | 0.739 0.558 |
| | avg | 0.269 0.292 | 0.242 0.267 | 0.238 **0.261** | 0.260 0.275 | **0.234** 0.273 | 0.238 0.275 | 0.241 0.283 | 0.242 0.279 | 0.279 0.301 | 0.300 0.342 | 0.597 0.495 |
| **Average** | | **0.309** 0.345 | 0.327 0.357 | 0.310 **0.344** | 0.329 0.350 | 0.336 0.368 | 0.360 0.378 | 0.407 0.416 | 0.378 0.389 | 0.491 0.446 | 0.678 0.579 | 1.904 0.995 |
| **1st count** | | 32 | 0 | 10 | 8 | 10 | 6 | 0 | 0 | 0 | 0 | 0 |

Table 13 shows the full results of zero-shot/few-shot long-term forecasting performance. VISIONTS achieves the best results in most cases (32 out of 62), outperforming MOIRAI$_{Base}$ (10 out of 62) and MOIRAI$_{Large}$ (8 out of 62).

## B.3   COMPARISON OF TRADITIONAL METHODS

In addition to deep learning models, we also compare traditional methods, including ARIMA, ETS, and two methods that require periodicity as our VISIONTS: Seasonal Naïve (repeating the last period) and Seasonal Avg (similar to Seasonal Naïve but repeating the average of all periods in the look-back window). Due to the high computational cost of ARIMA and ETS, we only compare them on the

Table 14: Comparison of traditional zero-shot forecasting baselines.

| Method | | VISIONTS | | ETS | | ARIMA | | Seasonal Naïve | | Seasonal Avg | |
|---|---|---|---|---|---|---|---|---|---|---|---|
| Metric | | MSE | MAE | MSE | MAE | MSE | MAE | MSE | MAE | MSE | MAE |
| *ETTh1* | 96 | **0.353** | **0.383** | 1.289 | 0.710 | 0.900 | 0.719 | 0.512 | 0.433 | 0.589 | 0.585 |
| | 192 | **0.392** | **0.410** | 1.319 | 0.730 | 0.906 | 0.724 | 0.581 | 0.469 | 0.598 | 0.590 |
| | 336 | **0.407** | **0.423** | 1.324 | 0.742 | 0.908 | 0.731 | 0.650 | 0.501 | 0.610 | 0.597 |
| | 720 | **0.406** | **0.441** | 1.329 | 0.751 | 0.932 | 0.753 | 0.655 | 0.514 | 0.656 | 0.624 |
| | avg | **0.390** | **0.414** | 1.315 | 0.733 | 0.912 | 0.732 | 0.600 | 0.479 | 0.613 | 0.599 |
| *ETTh2* | 96 | **0.271** | **0.328** | 0.399 | 0.408 | 0.488 | 0.508 | 0.391 | 0.380 | 0.457 | 0.494 |
| | 192 | **0.328** | **0.367** | 0.500 | 0.459 | 0.497 | 0.514 | 0.482 | 0.429 | 0.466 | 0.500 |
| | 336 | **0.345** | **0.381** | 0.562 | 0.498 | 0.507 | 0.522 | 0.532 | 0.466 | 0.476 | 0.509 |
| | 720 | **0.388** | **0.422** | 0.558 | 0.506 | 0.572 | 0.557 | 0.525 | 0.474 | 0.542 | 0.548 |
| | avg | **0.333** | **0.375** | 0.505 | 0.468 | 0.516 | 0.525 | 0.483 | 0.437 | 0.485 | 0.513 |
| *ETTm1* | 96 | **0.341** | **0.347** | 1.204 | 0.659 | 0.702 | 0.568 | 0.423 | 0.387 | 0.369 | 0.399 |
| | 192 | **0.360** | **0.360** | 1.251 | 0.685 | 0.704 | 0.570 | 0.463 | 0.406 | 0.374 | 0.402 |
| | 336 | **0.377** | **0.374** | 1.276 | 0.702 | 0.709 | 0.574 | 0.496 | 0.426 | 0.382 | 0.407 |
| | 720 | **0.416** | **0.405** | 1.311 | 0.724 | 0.713 | 0.580 | 0.574 | 0.464 | **0.394** | 0.416 |
| | avg | **0.374** | **0.372** | 1.261 | 0.693 | 0.707 | 0.573 | 0.489 | 0.421 | 0.380 | 0.406 |
| *ETTm2* | 96 | **0.228** | **0.282** | 0.257 | 0.324 | 0.397 | 0.434 | 0.263 | 0.301 | 0.365 | 0.411 |
| | 192 | **0.262** | **0.305** | 0.331 | 0.366 | 0.402 | 0.436 | 0.321 | 0.337 | 0.369 | 0.414 |
| | 336 | **0.293** | **0.328** | 0.402 | 0.406 | 0.407 | 0.439 | 0.376 | 0.370 | 0.375 | 0.418 |
| | 720 | **0.343** | **0.370** | 0.512 | 0.462 | 0.413 | 0.443 | 0.471 | 0.422 | 0.380 | 0.423 |
| | avg | **0.282** | **0.321** | 0.376 | 0.390 | 0.405 | 0.438 | 0.358 | 0.357 | 0.372 | 0.417 |
| **Average** | | **0.344** | **0.370** | 0.864 | 0.571 | 0.635 | 0.567 | 0.482 | 0.424 | 0.463 | 0.484 |
| **1st count** | | **41** | | 0 | | 0 | | 0 | | 1 | |

Table 15: Full results of Fig. 4: Forecasting results (MAE) on the Monash TSF benchmark. We reported the reproduction results of LLMTime based on the GPT3.5 API from Woo et al. (2024).

| | VISIONTS | LLMTime | MOIRAI$_{Small}$ | Naive | SES | Theta | TBATS | ETS | (DHR-)ARIMA | PR | CatBoost | FFNN | DeepAR | N-BEATS | WaveNet | Transformer |
|---|---|---|---|---|---|---|---|---|---|---|---|---|---|---|---|---|
| M1 Monthly | 1987.69 | 2562.84 | 2082.26 | 2707.75 | 2259.04 | 2166.18 | 2237.5 | 1905.28 | 2080.13 | 2088.25 | 2052.32 | 2162.58 | 1860.81 | **1820.37** | 2184.42 | 2723.88 |
| M3 Monthly | 737.93 | 877.97 | 713.41 | 837.14 | 743.41 | **623.71** | 630.59 | 626.46 | 654.8 | 692.97 | 732 | 692.48 | 728.81 | 648.6 | 699.3 | 798.38 |
| M3 Other | 315.85 | 300.3 | 263.54 | 278.43 | 277.83 | 215.35 | **189.42** | 194.98 | 193.02 | 234.43 | 318.13 | 240.17 | 247.56 | 221.85 | 245.29 | 239.24 |
| M4 Monthly | 666.54 | 728.27 | 597.6 | 671.27 | 625.24 | **563.58** | 589.52 | 582.6 | 575.36 | 596.19 | 611.69 | 612.52 | 615.22 | 578.48 | 655.51 | 780.47 |
| M4 Weekly | 404.23 | 518.44 | 339.76 | 347.99 | 336.82 | 333.32 | 296.15 | 335.66 | 321.61 | 293.21 | 364.65 | 338.37 | 351.78 | **277.73** | 359.46 | 378.89 |
| M4 Daily | 215.63 | 266.52 | 189.1 | 180.83 | 178.27 | 178.86 | **176.6** | 193.26 | 179.67 | 181.92 | 231.36 | 177.91 | 299.79 | 190.44 | 189.47 | 201.08 |
| M4 Hourly | 288.37 | 576.06 | 268.04 | 1218.06 | 1218.06 | 1220.97 | 386.27 | 3358.1 | 1310.85 | **257.39** | 285.35 | 385.49 | 886.02 | 425.75 | 393.63 | 320.54 |
| Tourism Quarterly | 12931.88 | 16918.86 | 18352.44 | 15845.1 | 15014.19 | **7656.49** | 9972.42 | 8925.52 | 10475.47 | 9092.58 | 10267.97 | 8981.04 | 9511.37 | 8640.56 | 9137.12 | 9521.67 |
| Tourism Monthly | 2560.19 | 5608.61 | 3569.85 | 5636.83 | 5302.1 | 2069.96 | 2940.08 | **2004.51** | 2536.77 | 2187.28 | 2537.04 | 2022.21 | 2003.02 | 2095.13 | 2146.98 |
| CIF 2016 | 570907.24 | 599313.8 | 655888.58 | 578596.5 | 581875.97 | 714818.6 | 855578.4 | 642421.4 | **469059** | 563205.57 | 603551.3 | 1495923 | 3200418 | 679034.8 | 5998225 | 407973 |
| Aus. Elec. Demand | 237.44 | 760.81 | 266.57 | 659.6 | 659.6 | 665.04 | 370.74 | 1282.99 | 1045.92 | 247.18 | 241.77 | 258.76 | 302.41 | **213.83** | 227.5 | 231.45 |
| Bitcoin | 2.33E+18 | 1.74E+18 | 1.76E+18 | 7.78E+17 | 5.33E+18 | 5.33E+18 | 9.9E+17 | 1.1E+18 | 3.62E+18 | **6.66E+17** | 1.93E+18 | 1.45E+18 | 1.95E+18 | 1.06E+18 | 2.46E+18 | 2.61E+18 |
| Pedestrian Counts | 52.01 | 97.77 | 54.88 | 170.88 | 170.87 | 170.94 | 222.38 | 216.5 | 635.16 | 44.18 | **43.41** | 46.41 | 44.78 | 66.84 | 46.46 | 47.29 |
| Vehicle Trips | 22.08 | 31.48 | 24.46 | 31.42 | 29.98 | 30.76 | **21.21** | 30.95 | 30.07 | 27.24 | 22.61 | 22.93 | 22 | 28.16 | 24.15 | 28.01 |
| KDD cup | 38.16 | 42.72 | 39.81 | 42.13 | 42.04 | 42.06 | 39.2 | 44.88 | 52.2 | 36.85 | **34.82** | 37.16 | 48.98 | 49.1 | 37.08 | 44.46 |
| Weather | 2.06 | 2.17 | **1.96** | 2.36 | 2.24 | 2.51 | 2.3 | 2.35 | 2.45 | 8.17 | 2.51 | 2.09 | 2.02 | 2.34 | 2.29 | 2.03 |
| NN5 Daily | **3.51** | 7.1 | 5.37 | 8.26 | 6.63 | 3.8 | 3.7 | 3.72 | 4.41 | 5.47 | 4.22 | 4.06 | 3.94 | 4.92 | 3.97 | 4.16 |
| NN5 Weekly | 14.67 | 15.76 | 15.07 | 16.71 | 15.66 | 15.3 | 14.98 | 15.7 | 15.38 | 14.94 | 15.29 | 15.02 | 14.69 | **14.19** | 19.34 | 20.34 |
| Carparts | 0.58 | 0.44 | 0.53 | 0.65 | 0.55 | 0.53 | 0.58 | 0.56 | 0.56 | 0.41 | 0.53 | **0.39** | **0.39** | 0.98 | 0.4 | **0.39** |
| FRED-MD | **1893.67** | 2804.64 | 2568.48 | 2825.67 | 2798.22 | 3492.84 | 1989.97 | 2041.42 | 2957.11 | 8921.94 | 2475.68 | 2339.57 | 4264.36 | 2557.8 | 2508.4 | 4666.04 |
| Traffic Hourly | **0.01** | 0.03 | 0.02 | 0.03 | 0.03 | 0.03 | 0.04 | 0.03 | 0.04 | 0.02 | 0.02 | **0.01** | **0.01** | 0.02 | 0.02 | **0.01** |
| Traffic Weekly | 1.14 | 1.15 | 1.17 | 1.19 | 1.12 | 1.13 | 1.17 | 1.14 | 1.22 | 1.13 | 1.17 | 1.15 | 1.18 | **1.11** | 1.2 | 1.42 |
| Rideshare | 5.92 | 6.28 | **1.35** | 6.29 | 6.29 | 7.62 | 6.45 | 6.29 | 3.37 | 6.3 | 6.07 | 6.59 | 6.28 | 5.55 | 2.75 | 6.29 |
| Hospital | 19.36 | 25.68 | 23 | 24.07 | 21.76 | 18.54 | **17.43** | 17.97 | 19.6 | 19.24 | 22.86 | 18.25 | 20.18 | 19.35 | | 36.19 |
| COVID Deaths | 137.51 | 653.31 | 124.32 | 353.71 | 353.71 | 321.32 | 96.29 | **85.59** | 85.77 | 347.98 | 475.15 | 144.14 | 201.98 | 158.81 | 1049.48 | 408.66 |
| Temperature Rain | 6.37 | 6.37 | 5.8 | 9.39 | 8.18 | 8.22 | 7.14 | 8.21 | 7.19 | 6.13 | 6.76 | 5.56 | 5.37 | 7.28 | 5.81 | **5.24** |
| Sunspot | 2.81 | 5.07 | **0.11** | 3.93 | 4.93 | 4.93 | 2.57 | 4.93 | 2.57 | 3.83 | 2.27 | 7.97 | 0.77 | 14.47 | 0.17 | 0.13 |
| Saugeen River Flow | 30.22 | 34.84 | 24.07 | 21.5 | 21.5 | 21.49 | 22.26 | 30.69 | 22.38 | 25.24 | **21.28** | 22.98 | 23.51 | 27.92 | 22.17 | 28.06 |
| US Births | 519.94 | 1374.99 | 872.51 | 1152.67 | 1192.2 | 586.93 | **399** | 419.73 | 526.33 | 574.93 | 441.7 | 557.87 | 424.93 | 422 | 504.4 | 452.87 |
| **Normalized MAE** | 0.729 | 1.041 | 0.657 | 1.000 | 1.028 | 0.927 | 0.758 | 0.872 | 0.898 | 0.785 | 0.760 | 0.741 | 0.759 | 0.783 | 0.749 | 0.770 |
| **Rank** | 2 | 16 | 1 | 14 | 15 | 13 | 5 | 11 | 12 | 10 | 7 | 3 | 6 | 9 | 4 | 8 |

small-scale benchmarks, *i.e.*, four ETT datasets. Table 14 shows that VISIONTS also achieves the best performance.

### B.4 FULL FORECASTING RESULTS OF THE MONASH TSF BENCHMARK

**Setup** Table 10 lists the sampling frequency and the selected period $P$ for each dataset. Datasets with $P = 1$ indicate no significant periodicity, where we use a context length of $L = 300$. For other datasets with $P > 1$, we select a longer context length of $L = 1000$. All datasets were tested with the hyperparameters $r = c = 0.4$ as we had done for the long-term TSF benchmark.

**Results** Table 15 presents VISIONTS 's MAE test results, with the normalized MAE calculated by dividing each dataset's MAE by the naive forecast's MAE and aggregated using the geometric mean across datasets. We include the result of each baseline from Woo et al. (2024). Particularly, we find that VISIONTS outperforms MOIRAI on some datasets with $P = 1$ (*e.g.*, FRED-MD and NN5 Weekly), showing that VISIONTS can still work effectively without significant periodicity.

Table 16: Results on the PF benchmark. Results of baselines are based on Woo et al. (2024).

| | | Zero-Shot | | | | Full-Shot | | | | Baseline | |
| | | VISIONTS | MOIRAI$_{Small}$ | MOIRAI$_{Base}$ | MOIRAI$_{Large}$ | PatchTST | TiDE | TFT | DeepAR | AutoARIMA | Seasonal Naïve |
|---|---|---|---|---|---|---|---|---|---|---|---|
| Electricity | sMAPE | 0.109 | 0.134 | 0.111 | 0.106 | 0.107 | **0.102** | 0.106 | 0.118 | 0.318 | 0.108 |
| | MASE | 0.755 | 0.981 | 0.792 | 0.751 | 0.753 | **0.706** | 0.747 | 0.844 | 3.229 | 0.881 |
| | ND | **0.061** | 0.092 | 0.069 | 0.063 | 0.065 | **0.061** | 0.063 | 0.080 | 0.357 | 0.070 |
| | NRMSE | **0.448** | 0.840 | 0.551 | 0.465 | 0.506 | 0.514 | 0.511 | 0.704 | 3.296 | 0.478 |
| Solar | sMAPE | 1.370 | 1.445 | 1.410 | 1.400 | 1.501 | 1.400 | 1.391 | 1.385 | 1.685 | **0.691** |
| | MASE | **1.141** | 1.465 | 1.292 | 1.237 | 1.607 | 1.265 | 1.399 | 1.222 | 2.583 | 1.203 |
| | ND | **0.484** | 0.624 | 0.551 | 0.528 | 0.685 | 0.538 | 0.594 | 0.520 | 1.098 | 0.512 |
| | NRMSE | **0.975** | 1.135 | 1.034 | 1.014 | 1.408 | 1.093 | 1.236 | 1.033 | 1.784 | 1.168 |
| Walmart | sMAPE | 0.167 | 0.179 | 0.168 | 0.174 | 0.150 | **0.145** | 0.172 | 0.216 | 0.219 | 0.205 |
| | MASE | 0.949 | 1.048 | 0.964 | 1.007 | 0.867 | **0.814** | 0.948 | 1.193 | 1.131 | 1.236 |
| | ND | 0.108 | 0.129 | 0.117 | 0.124 | 0.105 | **0.097** | 0.108 | 0.147 | 0.141 | 0.151 |
| | NRMSE | 0.225 | 0.324 | 0.291 | 0.332 | 0.218 | **0.204** | 0.235 | 0.298 | 0.305 | 0.328 |
| Weather | sMAPE | 0.672 | 0.686 | 0.623 | 0.688 | 0.668 | 0.636 | 0.672 | 0.776 | 0.770 | **0.401** |
| | MASE | 0.737 | 0.521 | **0.487** | 0.515 | 0.844 | 0.832 | 0.692 | 3.170 | 0.938 | 0.782 |
| | ND | 0.063 | 0.063 | **0.048** | 0.063 | 0.072 | 0.066 | 0.051 | 0.163 | 0.139 | 0.068 |
| | NRMSE | 0.247 | 0.229 | 0.417 | 0.331 | 0.260 | 0.214 | **0.211** | 0.486 | 0.465 | 0.290 |
| Istanbul Traffic | sMAPE | **0.243** | 0.359 | 0.284 | 0.288 | 0.287 | 0.280 | 0.287 | 0.249 | 1.141 | 0.391 |
| | MASE | 0.706 | 0.990 | 0.644 | 0.631 | 0.653 | 0.618 | 0.620 | **0.613** | 3.358 | 1.137 |
| | ND | 0.160 | 0.224 | 0.146 | 0.143 | 0.148 | 0.140 | 0.141 | **0.139** | 0.758 | 0.257 |
| | NRMSE | 0.250 | 0.294 | 0.194 | 0.186 | 0.190 | 0.185 | 0.185 | **0.181** | 0.959 | 0.384 |
| Turkey Power | sMAPE | 0.386 | 0.389 | 0.378 | 0.375 | 0.416 | 0.389 | 0.383 | 0.404 | 0.244 | **0.125** |
| | MASE | **0.856** | 0.948 | 0.888 | 0.870 | 1.234 | 0.904 | 0.890 | 1.395 | 1.700 | 0.906 |
| | ND | 0.062 | 0.061 | 0.051 | **0.046** | 0.071 | 0.059 | 0.049 | 0.083 | 0.150 | 0.085 |
| | NRMSE | 0.154 | 0.149 | 0.118 | **0.102** | 0.158 | 0.139 | 0.104 | 0.181 | 0.383 | 0.231 |
| **1$^{st}$ count** | | **7** | 0 | 2 | 2 | 0 | 7 | 0 | 3 | 0 | 3 |

## B.5 FULL FORECASTING RESULTS OF THE PF BENCHMARK

Table 17: Comparison of LaMa as the backbone. Results are averaged on four prediction lengths.

| | MAE | | LaMa | | MOIRAI$_{Small}$ | | MOIRAI$_{Large}$ | |
| | MSE | MAE | MSE | MAE | MSE | MAE | MSE | MAE |
|---|---|---|---|---|---|---|---|---|
| ETTh1 | 0.390 | 0.414 | 0.425 | 0.433 | 0.400 | 0.424 | 0.510 | 0.469 |
| ETTh2 | 0.333 | 0.375 | 0.376 | 0.408 | 0.341 | 0.379 | 0.354 | 0.377 |
| ETTm1 | 0.374 | 0.372 | 0.400 | 0.391 | 0.448 | 0.410 | 0.390 | 0.389 |
| ETTm2 | 0.282 | 0.321 | 0.294 | 0.337 | 0.300 | 0.341 | 0.276 | 0.320 |
| **Average** | 0.344 | 0.370 | 0.374 | 0.392 | 0.372 | 0.388 | 0.382 | 0.388 |

For all datasets on the PF benchmark, we use $c = r = 0.4$, and a context length $L = 2000$. Table 16 summarizes the results on the PF benchmark, where our VISIONTS outperforms MOIRAI in the zero-shot setting and is comparable with the best full-shot method, TiDE.

## B.6 IMPACT OF BACKBONES

Table 18 compares zero-shot forecasting performance of three MAE variants (112M, 330M, and 657M), showing that the three variants are similar, but larger models show a slight decrease. Particularly, the smallest model excels in ETTh2, ETTm1, ETTm2, and Weather, while the largest model excels in Electricity. Additionally, Table 17 compares VISIONTS with another visual backbone, LaMa.

## B.7 IMPACT OF THE DIFFERENT IMAGE ENCODING STRATEGIES

Table 19 summarizes the impact of interpolation strategies and image orientations in the Alignment step. It shows that the smoother Bilinear and Bicubic interpolation perform similarly, both significantly better than the rougher Nearest Neighbor. This suggests that smooth resizing effectively handles time series interpolation. Moreover, image orientation has little impact on performance.

## B.8 HYPERPARAMETER ANALYSIS

Figs. 7 to 9 show the influence of three hyperparameters, $r$, $c$, and $L$. We report the MSE averaged on four prediction lengths $\{96, 192, 336, 720\}$.

Table 18: Full results of Table 4: zero-shot forecasting results of different MAE variants. **Bold**: best results among three variants. We also include the results from MOIRAI for reference.

| Method | | MAE (Base) 112M | | MAE (Large) 330M | | MAE (Huge) 657M | | MOIRAI (Small) 14M | | MOIRAI (Base) 91M | | MOIRAI (Huge) 311M | |
|---|---|---|---|---|---|---|---|---|---|---|---|---|---|
| Metric | | MSE | MAE | MSE | MAE | MSE | MAE | MSE | MAE | MSE | MAE | MSE | MAE |
| ETTh1 | 96 | 0.353 | 0.383 | **0.346** | **0.382** | 0.362 | 0.384 | *0.375* | *0.402* | *0.384* | *0.402* | *0.380* | *0.398* |
| | 192 | 0.392 | 0.410 | **0.379** | **0.406** | 0.407 | 0.414 | *0.399* | *0.419* | *0.425* | *0.429* | *0.440* | *0.434* |
| | 336 | 0.407 | 0.423 | **0.391** | **0.416** | 0.399 | 0.419 | *0.412* | *0.429* | *0.456* | *0.450* | *0.514* | *0.474* |
| | 720 | 0.406 | 0.441 | 0.397 | **0.433** | **0.395** | **0.433** | *0.413* | *0.444* | *0.470* | *0.473* | *0.705* | *0.568* |
| | avg | 0.390 | 0.414 | **0.378** | **0.409** | 0.391 | 0.412 | *0.400* | *0.424* | *0.434* | *0.439* | *0.510* | *0.469* |
| ETTh2 | 96 | **0.271** | **0.328** | 0.286 | 0.334 | 0.285 | 0.333 | *0.281* | *0.334* | *0.277* | *0.327* | *0.287* | *0.325* |
| | 192 | **0.328** | **0.367** | 0.346 | 0.375 | 0.337 | 0.369 | *0.340* | *0.373* | *0.340* | *0.374* | *0.347* | *0.367* |
| | 336 | **0.345** | **0.381** | 0.356 | 0.387 | 0.357 | 0.388 | *0.362* | *0.393* | *0.371* | *0.401* | *0.377* | *0.393* |
| | 720 | 0.388 | 0.422 | **0.371** | **0.409** | 0.379 | 0.412 | *0.380* | *0.416* | *0.394* | *0.426* | *0.404* | *0.421* |
| | avg | **0.333** | **0.375** | 0.340 | 0.377 | 0.339 | **0.375** | *0.341* | *0.379* | *0.346* | *0.382* | *0.354* | *0.377* |
| ETTm1 | 96 | **0.341** | **0.347** | 0.344 | 0.349 | 0.352 | 0.351 | *0.404* | *0.383* | *0.335* | *0.360* | *0.353* | *0.363* |
| | 192 | **0.360** | **0.360** | 0.365 | 0.363 | **0.360** | 0.367 | *0.435* | *0.402* | *0.366* | *0.379* | *0.376* | *0.380* |
| | 336 | **0.377** | **0.374** | 0.381 | 0.376 | 0.381 | 0.383 | *0.462* | *0.416* | *0.391* | *0.394* | *0.399* | *0.395* |
| | 720 | **0.416** | **0.405** | 0.429 | 0.411 | 0.440 | 0.412 | *0.490* | *0.437* | *0.434* | *0.419* | *0.432* | *0.417* |
| | avg | **0.374** | **0.372** | 0.379 | 0.375 | 0.383 | 0.378 | *0.448* | *0.410* | *0.382* | *0.388* | *0.390* | *0.389* |
| ETTm2 | 96 | 0.228 | **0.282** | **0.225** | **0.282** | 0.229 | **0.282** | *0.205* | *0.282* | *0.195* | *0.269* | *0.189* | *0.260* |
| | 192 | **0.262** | **0.305** | **0.262** | **0.305** | 0.265 | 0.306 | *0.261* | *0.318* | *0.247* | *0.303* | *0.247* | *0.300* |
| | 336 | 0.293 | 0.328 | 0.299 | 0.331 | **0.286** | **0.324** | *0.319* | *0.355* | *0.291* | *0.333* | *0.295* | *0.334* |
| | 720 | **0.343** | **0.370** | 0.358 | 0.377 | 0.355 | 0.374 | *0.415* | *0.410* | *0.355* | *0.377* | *0.372* | *0.386* |
| | avg | **0.282** | **0.321** | 0.286 | 0.324 | 0.284 | 0.322 | *0.300* | *0.341* | *0.272* | *0.321* | *0.276* | *0.320* |
| Electricity | 96 | 0.177 | 0.266 | 0.177 | 0.268 | **0.170** | **0.259** | *0.205* | *0.299* | *0.158* | *0.248* | *0.152* | *0.242* |
| | 192 | 0.188 | 0.277 | 0.192 | 0.283 | **0.182** | **0.273** | *0.220* | *0.310* | *0.174* | *0.263* | *0.171* | *0.259* |
| | 336 | **0.207** | 0.296 | 0.213 | 0.303 | **0.207** | **0.295** | *0.236* | *0.323* | *0.191* | *0.278* | *0.192* | *0.278* |
| | 720 | 0.256 | 0.337 | 0.256 | 0.337 | **0.250** | **0.333** | *0.270* | *0.347* | *0.229* | *0.307* | *0.236* | *0.313* |
| | avg | 0.207 | 0.294 | 0.209 | 0.298 | **0.202** | **0.290** | *0.233* | *0.320* | *0.188* | *0.274* | *0.188* | *0.273* |
| Weather | 96 | **0.220** | **0.257** | 0.222 | **0.257** | 0.235 | 0.265 | *0.173* | *0.212* | *0.167* | *0.203* | *0.177* | *0.208* |
| | 192 | **0.244** | **0.275** | 0.246 | **0.275** | 0.276 | 0.288 | *0.216* | *0.250* | *0.209* | *0.241* | *0.219* | *0.249* |
| | 336 | **0.280** | **0.299** | 0.283 | 0.301 | 0.304 | 0.309 | *0.260* | *0.282* | *0.256* | *0.276* | *0.277* | *0.292* |
| | 720 | **0.330** | **0.337** | 0.338 | 0.343 | 0.351 | 0.350 | *0.320* | *0.322* | *0.321* | *0.323* | *0.365* | *0.350* |
| | avg | **0.269** | **0.292** | 0.272 | 0.294 | 0.292 | 0.303 | *0.242* | *0.267* | *0.238* | *0.261* | *0.260* | *0.275* |
| **Average** | | **0.309** | **0.345** | 0.311 | 0.346 | 0.315 | 0.347 | *0.327* | *0.357* | *0.310* | *0.344* | *0.329* | *0.350* |
| **1st count** | | 38 | | 17 | | 17 | | - | | - | | - | |

Table 19: Impact of resampling filters and image orientations.

| Method | | Interpolation strategies in resampling | | | | | |
|---|---|---|---|---|---|---|---|
| | | Bilinear | | Bicubic | | Nearest Neighbor | |
| Metric | | MSE | MAE | MSE | MAE | MSE | MAE |
| ETTh1 | 96 | 0.353 | **0.383** | **0.351** | **0.383** | 0.426 | 0.424 |
| | 192 | **0.392** | 0.410 | **0.392** | **0.409** | 0.450 | 0.443 |
| | 336 | **0.407** | 0.423 | **0.407** | **0.422** | 0.451 | 0.450 |
| | 720 | 0.406 | 0.441 | **0.405** | **0.440** | 0.454 | 0.470 |
| | avg | 0.390 | **0.414** | 0.389 | **0.414** | 0.445 | 0.446 |
| ETTh2 | 96 | **0.271** | **0.328** | 0.274 | 0.329 | 0.298 | 0.349 |
| | 192 | **0.328** | **0.367** | 0.330 | **0.367** | 0.343 | 0.380 |
| | 336 | **0.345** | 0.381 | **0.345** | **0.380** | 0.373 | 0.401 |
| | 720 | 0.388 | 0.422 | **0.386** | **0.419** | 0.404 | 0.431 |
| | avg | **0.333** | 0.375 | 0.334 | **0.374** | 0.354 | 0.390 |
| ETTm1 | 96 | **0.341** | **0.347** | 0.366 | 0.354 | 0.399 | 0.374 |
| | 192 | **0.360** | **0.360** | 0.383 | 0.367 | 0.397 | 0.376 |
| | 336 | **0.377** | **0.374** | 0.396 | 0.381 | 0.386 | 0.380 |
| | 720 | **0.416** | **0.405** | 0.429 | 0.409 | 0.417 | 0.409 |
| | avg | **0.374** | **0.372** | 0.393 | 0.378 | 0.400 | 0.384 |
| ETTm2 | 96 | **0.228** | **0.282** | 0.246 | 0.296 | 0.264 | 0.326 |
| | 192 | **0.262** | **0.305** | 0.273 | 0.313 | 0.273 | 0.328 |
| | 336 | **0.293** | **0.328** | 0.303 | 0.334 | 0.297 | 0.343 |
| | 720 | 0.343 | 0.370 | 0.343 | 0.370 | **0.334** | **0.369** |
| | avg | **0.282** | **0.321** | 0.291 | 0.328 | 0.292 | 0.341 |
| **Average** | | **0.344** | **0.370** | 0.352 | 0.373 | 0.373 | 0.391 |
| **1st count** | | 30 | | 18 | | 2 | |

| Method | | Image orientation | | | | | |
|---|---|---|---|---|---|---|---|
| | | - | | Horizontal flip | | Vertical flip | |
| Metric | | MSE | MAE | MSE | MAE | MSE | MAE |
| ETTh1 | 96 | 0.353 | 0.383 | **0.348** | **0.379** | 0.355 | 0.385 |
| | 192 | 0.392 | 0.410 | **0.386** | **0.404** | 0.394 | 0.411 |
| | 336 | 0.407 | 0.423 | **0.401** | **0.416** | 0.408 | 0.423 |
| | 720 | 0.406 | 0.441 | **0.399** | **0.430** | 0.406 | 0.442 |
| | avg | 0.390 | 0.414 | **0.384** | **0.407** | 0.391 | 0.415 |
| ETTh2 | 96 | **0.271** | **0.328** | 0.274 | 0.329 | 0.274 | 0.330 |
| | 192 | **0.328** | **0.367** | 0.331 | 0.370 | 0.330 | **0.367** |
| | 336 | **0.345** | **0.381** | 0.347 | 0.386 | **0.345** | **0.381** |
| | 720 | 0.388 | 0.422 | **0.376** | **0.416** | 0.388 | 0.422 |
| | avg | 0.333 | **0.375** | **0.332** | 0.375 | 0.334 | **0.375** |
| ETTm1 | 96 | **0.341** | **0.347** | 0.345 | 0.348 | 0.342 | **0.347** |
| | 192 | **0.360** | **0.360** | 0.364 | 0.362 | **0.360** | **0.360** |
| | 336 | **0.377** | **0.374** | 0.378 | 0.375 | **0.377** | **0.374** |
| | 720 | **0.416** | **0.405** | 0.419 | 0.408 | 0.417 | **0.405** |
| | avg | **0.374** | **0.372** | 0.376 | 0.373 | **0.374** | **0.372** |
| ETTm2 | 96 | **0.228** | **0.282** | 0.230 | 0.286 | **0.228** | 0.283 |
| | 192 | **0.262** | **0.305** | 0.264 | 0.308 | **0.262** | **0.305** |
| | 336 | **0.293** | **0.328** | 0.298 | 0.332 | **0.293** | **0.328** |
| | 720 | **0.343** | 0.370 | 0.350 | 0.373 | **0.343** | **0.369** |
| | avg | **0.282** | **0.321** | 0.285 | 0.325 | **0.282** | **0.321** |
| **Average** | | **0.344** | **0.370** | 0.344 | 0.370 | 0.345 | 0.371 |
| **1st count** | | 28 | | 16 | | 21 | |

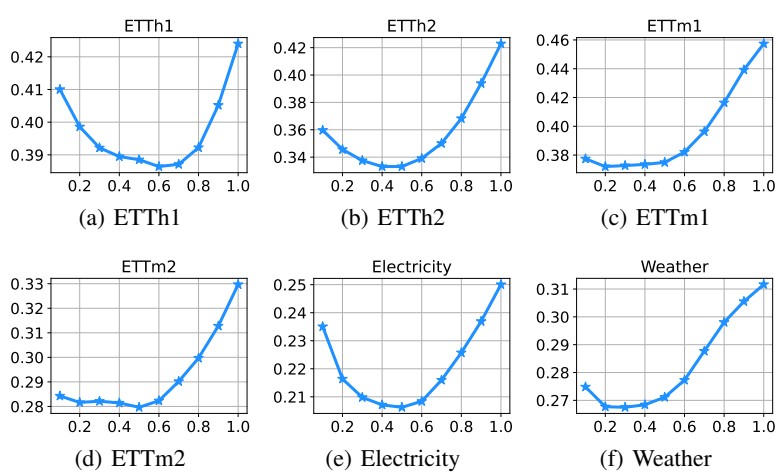

Figure 7: MSE (Y-axis) performance of different normalization constants $r$ (X-axis).

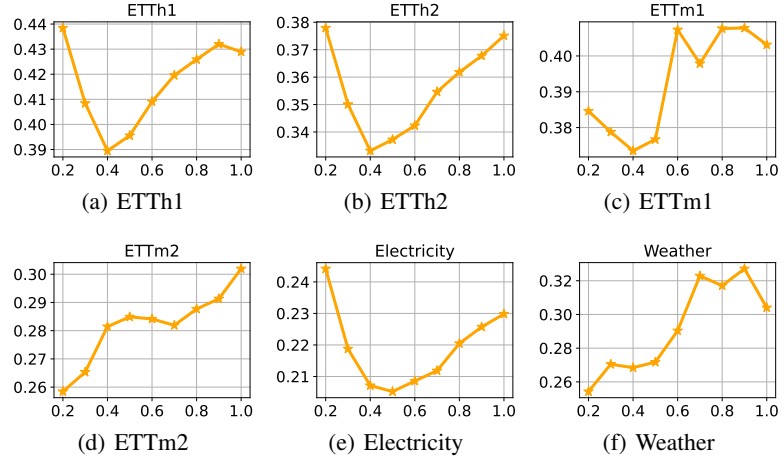

Figure 8: MSE (Y-axis) performance of different alignment constants $c$ (X-axis).

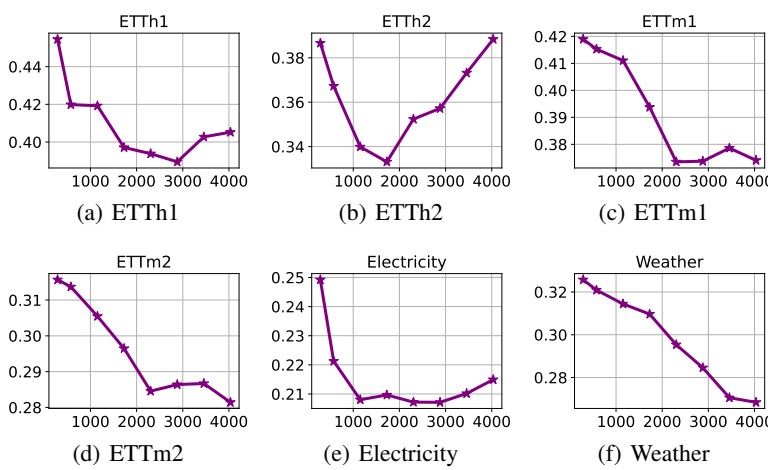

Figure 9: MSE (Y-axis) performance of different context lengths $L$ (X-axis).

# C    FULL-SHOT FORECASTING

## C.1    TRAINING DETAILS

Table 20: Final hyperparameters for VISIONTS used in our full-shot forecasting.

|  | ETTh1 | ETTh2 | ETTm1 | ETTm2 | Illness | Weather | Traffic | Electricity |
|---|---|---|---|---|---|---|---|---|
| Normalization constant $r$ | 0.4 | 0.4 | 0.4 | 0.4 | 1.0 | 1.0 | 0.4 | 0.4 |
| Alignment constant $c$ | 0.4 | 0.4 | 0.4 | 0.4 | 0.4 | 0.7 | 0.4 | 0.4 |
| Context length $L$ | 1152 | 1152 | 2304 | 1152 | 104 | 576 | 1152 | 1152 |

Based on the principle of channel independence (Nie et al., 2022; Han et al., 2024), we treat the variables of each time series as individual data samples. We use an Adam optimizer with a learning rate 0.0001 and a batch size 256 to fine-tune MAE. All experiments are repeated three times. The training epoch is one for all the datasets except Illness, for which we train MAE for 100 epochs with an early stop due to the limited training dataset scale. We conduct tuning on validation sets for the three hyperparameters, $r$, $c$, and $L$. The final hyperparameters used are summarized in Table 20.

## C.2    STANDARD DEVIATIONS

Table 21: Standard deviations of full-shot experiments.

| Method | | VISIONTS | | Time-LLM | | GPT4TS | |
|---|---|---|---|---|---|---|---|
| Metric | | MSE | MAE | MSE | MAE | MSE | MAE |
| ETTh1 | 96 | **0.347 ± 0.002** | **0.376 ± 0.000** | 0.376 ± 0.003 | 0.402 ± 0.002 | 0.370 ± 0.003 | 0.389 ± 0.001 |
|  | 192 | **0.385 ± 0.001** | **0.400 ± 0.000** | 0.407 ± 0.003 | 0.421 ± 0.002 | 0.412 ± 0.003 | 0.413 ± 0.001 |
|  | 336 | **0.407 ± 0.001** | **0.415 ± 0.001** | 0.430 ± 0.004 | 0.438 ± 0.001 | 0.448 ± 0.003 | 0.431 ± 0.001 |
|  | 720 | **0.439 ± 0.001** | **0.443 ± 0.000** | 0.457 ± 0.003 | 0.468 ± 0.001 | 0.441 ± 0.003 | 0.449 ± 0.001 |
| ETTh2 | 96 | **0.269 ± 0.003** | **0.328 ± 0.002** | 0.286 ± 0.003 | 0.346 ± 0.002 | 0.280 ± 0.001 | 0.335 ± 0.001 |
|  | 192 | **0.332 ± 0.001** | **0.374 ± 0.001** | 0.361 ± 0.003 | 0.391 ± 0.002 | 0.348 ± 0.002 | 0.380 ± 0.001 |
|  | 336 | **0.351 ± 0.002** | **0.395 ± 0.002** | 0.390 ± 0.003 | 0.414 ± 0.002 | 0.380 ± 0.002 | 0.405 ± 0.001 |
|  | 720 | **0.390 ± 0.003** | **0.430 ± 0.002** | 0.405 ± 0.003 | 0.434 ± 0.002 | 0.406 ± 0.002 | 0.436 ± 0.001 |
| ETTm1 | 96 | **0.281 ± 0.001** | **0.322 ± 0.001** | 0.291 ± 0.001 | 0.341 ± 0.001 | 0.300 ± 0.001 | 0.340 ± 0.000 |
|  | 192 | **0.322 ± 0.006** | **0.353 ± 0.002** | 0.341 ± 0.001 | 0.369 ± 0.001 | 0.343 ± 0.001 | 0.368 ± 0.000 |
|  | 336 | **0.356 ± 0.003** | **0.379 ± 0.002** | 0.359 ± 0.002 | 0.379 ± 0.001 | 0.376 ± 0.001 | 0.386 ± 0.000 |
|  | 720 | **0.391 ± 0.001** | **0.413 ± 0.001** | 0.433 ± 0.001 | 0.419 ± 0.001 | 0.431 ± 0.001 | 0.416 ± 0.000 |
| ETTm2 | 96 | 0.169 ± 0.003 | 0.256 ± 0.002 | **0.162 ± 0.001** | **0.248 ± 0.001** | 0.163 ± 0.001 | 0.249 ± 0.001 |
|  | 192 | 0.225 ± 0.003 | 0.294 ± 0.003 | 0.235 ± 0.002 | 0.304 ± 0.001 | **0.222 ± 0.001** | **0.291 ± 0.000** |
|  | 336 | 0.278 ± 0.002 | 0.334 ± 0.001 | 0.280 ± 0.002 | 0.329 ± 0.001 | **0.273 ± 0.001** | **0.327 ± 0.001** |
|  | 720 | 0.372 ± 0.002 | 0.392 ± 0.002 | 0.366 ± 0.002 | 0.382 ± 0.001 | **0.357 ± 0.001** | **0.376 ± 0.001** |
| Weather | 96 | **0.142 ± 0.000** | 0.192 ± 0.001 | 0.155 ± 0.001 | 0.199 ± 0.001 | 0.148 ± 0.001 | **0.188 ± 0.000** |
|  | 192 | **0.191 ± 0.000** | 0.238 ± 0.000 | 0.223 ± 0.001 | 0.261 ± 0.001 | 0.192 ± 0.001 | **0.230 ± 0.000** |
|  | 336 | **0.246 ± 0.003** | 0.282 ± 0.001 | 0.251 ± 0.001 | 0.279 ± 0.001 | **0.246 ± 0.001** | **0.273 ± 0.000** |
|  | 720 | 0.328 ± 0.004 | 0.337 ± 0.001 | 0.345 ± 0.001 | 0.342 ± 0.001 | **0.320 ± 0.001** | **0.328 ± 0.000** |
| Traffic | 96 | **0.344 ± 0.001** | **0.236 ± 0.000** | 0.392 ± 0.001 | 0.267 ± 0.000 | 0.396 ± 0.001 | 0.264 ± 0.000 |
|  | 192 | **0.372 ± 0.001** | **0.249 ± 0.001** | 0.409 ± 0.001 | 0.271 ± 0.000 | 0.412 ± 0.001 | 0.268 ± 0.000 |
|  | 336 | **0.383 ± 0.001** | **0.257 ± 0.001** | 0.434 ± 0.001 | 0.296 ± 0.000 | 0.421 ± 0.001 | 0.273 ± 0.000 |
|  | 720 | **0.422 ± 0.001** | **0.280 ± 0.000** | 0.451 ± 0.001 | 0.291 ± 0.000 | 0.455 ± 0.001 | 0.291 ± 0.000 |
| Electricity | 96 | **0.126 ± 0.000** | **0.218 ± 0.000** | 0.137 ± 0.000 | 0.233 ± 0.000 | 0.141 ± 0.000 | 0.239 ± 0.000 |
|  | 192 | **0.146 ± 0.001** | **0.239 ± 0.001** | 0.152 ± 0.000 | 0.247 ± 0.000 | 0.158 ± 0.000 | 0.253 ± 0.000 |
|  | 336 | **0.161 ± 0.001** | **0.255 ± 0.001** | 0.169 ± 0.000 | 0.267 ± 0.000 | 0.172 ± 0.000 | 0.266 ± 0.000 |
|  | 720 | **0.193 ± 0.000** | **0.286 ± 0.000** | 0.200 ± 0.000 | 0.290 ± 0.000 | 0.207 ± 0.000 | 0.293 ± 0.000 |
| 1st count | | **42** | | 2 | | 12 | |

We report the standard deviations of our full-shot experiments computed on three runs in Table 21, including the results of Time-LLM and GPT4TS from Tan et al. (2024) for reference.

### C.3 ABLATION STUDY AND FINE-TUNING STRATEGY COMPARISON

Table 22: Full results of Tables 7 and 8: Ablation studies (left) and fine-tuning strategies (right). Results are averaged on four prediction lengths: {96, 192, 336, 720}.

| | | Ablation on Visual MAE (VM) | | | | |
| --- | --- | --- | --- | --- | --- | --- |
| | | - | w/o VM | VM2Attn | VM2Trsf | Rand-VM |
| ETTh1 | MSE | **0.395** | 0.785 | 0.448 | 0.459 | 0.534 |
| | MAE | **0.409** | 0.649 | 0.458 | 0.462 | 0.470 |
| ETTh2 | MSE | **0.336** | 0.420 | 0.418 | 0.448 | 0.411 |
| | MAE | **0.382** | 0.453 | 0.445 | 0.457 | 0.432 |
| ETTm1 | MSE | **0.338** | 0.676 | 0.397 | 0.398 | 0.433 |
| | MAE | **0.367** | 0.562 | 0.415 | 0.410 | 0.413 |
| ETTm2 | MSE | **0.261** | 0.379 | 0.274 | 0.292 | 0.288 |
| | MAE | **0.319** | 0.415 | 0.334 | 0.344 | 0.341 |
| Average | MSE | **0.333** | 0.565 | 0.384 | 0.399 | 0.417 |
| | MAE | **0.369** | 0.520 | 0.413 | 0.418 | 0.414 |
| 1st count | | 10 | 0 | 0 | 0 | 0 |

| | | Ablation on trained parameters | | | | | |
| --- | --- | --- | --- | --- | --- | --- | --- |
| | | All | LN | Bias | MLP | Attn | Freeze |
| ETTh1 | MSE | 0.534 | **0.395** | 0.401 | 0.534 | 0.554 | 0.419 |
| | MAE | 0.470 | **0.409** | 0.414 | 0.471 | 0.479 | 0.418 |
| ETTh2 | MSE | 0.411 | **0.336** | 0.347 | 0.401 | 0.392 | 0.340 |
| | MAE | 0.432 | 0.382 | 0.392 | 0.419 | 0.414 | **0.376** |
| ETTm1 | MSE | 0.433 | **0.338** | 0.343 | 0.441 | 0.444 | 0.374 |
| | MAE | 0.413 | **0.367** | 0.368 | 0.415 | 0.415 | 0.372 |
| ETTm2 | MSE | 0.288 | 0.261 | **0.256** | 0.292 | 0.289 | 0.305 |
| | MAE | 0.341 | 0.319 | **0.318** | 0.342 | 0.339 | 0.334 |
| Average | MSE | 0.417 | **0.333** | 0.337 | 0.417 | 0.420 | 0.360 |
| | MAE | 0.414 | **0.369** | 0.373 | 0.412 | 0.412 | 0.375 |
| 1st count | | 0 | 7 | 2 | 0 | 0 | 1 |

We compare the following ablation variants to verify the role of the visual model (VM), similar to Tan et al. (2024).

- **w/o VM** removes all the transformer blocks in encoders and decoders.
- **VM2Attn** replaces both the encoder and decoder with a self-attention layer, matching MAE structure but with random initialization.
- **VM2Trsf** is similar to **VM2Attn** but replaces them with a Transformer block (*i.e.*, a self-attention layer plus an MLP layer).
- **Rand-VM** keeps the same architecture as the vanilla MAE, but all the weights are randomly initialized.

We also compare fine-tuning different components in MAE as follows:

- **All** fine-tunes all the trainable weights in MAE.
- **LN** fine-tunes only the layer normalization, which is the default setting used in our experiments.
- **Bias** fine-tunes only the bias term of all the linear layers, proposed by Zaken et al. (2022).
- **MLP** and **Attn** fine-tune only the feed-forward layer and the self-attention layer, respectively.
- **Freeze** does not fine-tune any weight. Note that it differs from the previous zero-shot experiment, where a longer context length was used (see Table 12 and Table 20).

The results are shown in Table 22, suggesting that visual knowledge is crucial for VISIONTS and fine-tuning the layer normalization is the best.

## D VISUALIZATION

We visualized the predictions of VISIONTS in the zero-shot setting, including its input and reconstructed images. We also visualized the predictions of MOIRAI_Large and Seasonal Naïve, with their MAE metrics for comparison. Figs. 10 to 12 show examples where VISIONTS performed well, with Fig. 10 depicting a more regular pattern, while Figs. 11 and 12 display less obvious patterns. Fig. 13 illustrates a case where VISIONTS underperformed, as it aggressively predicted the trend despite the lack of clear patterns in the input sequence, whereas MOIRAI_Large made more conservative predictions.

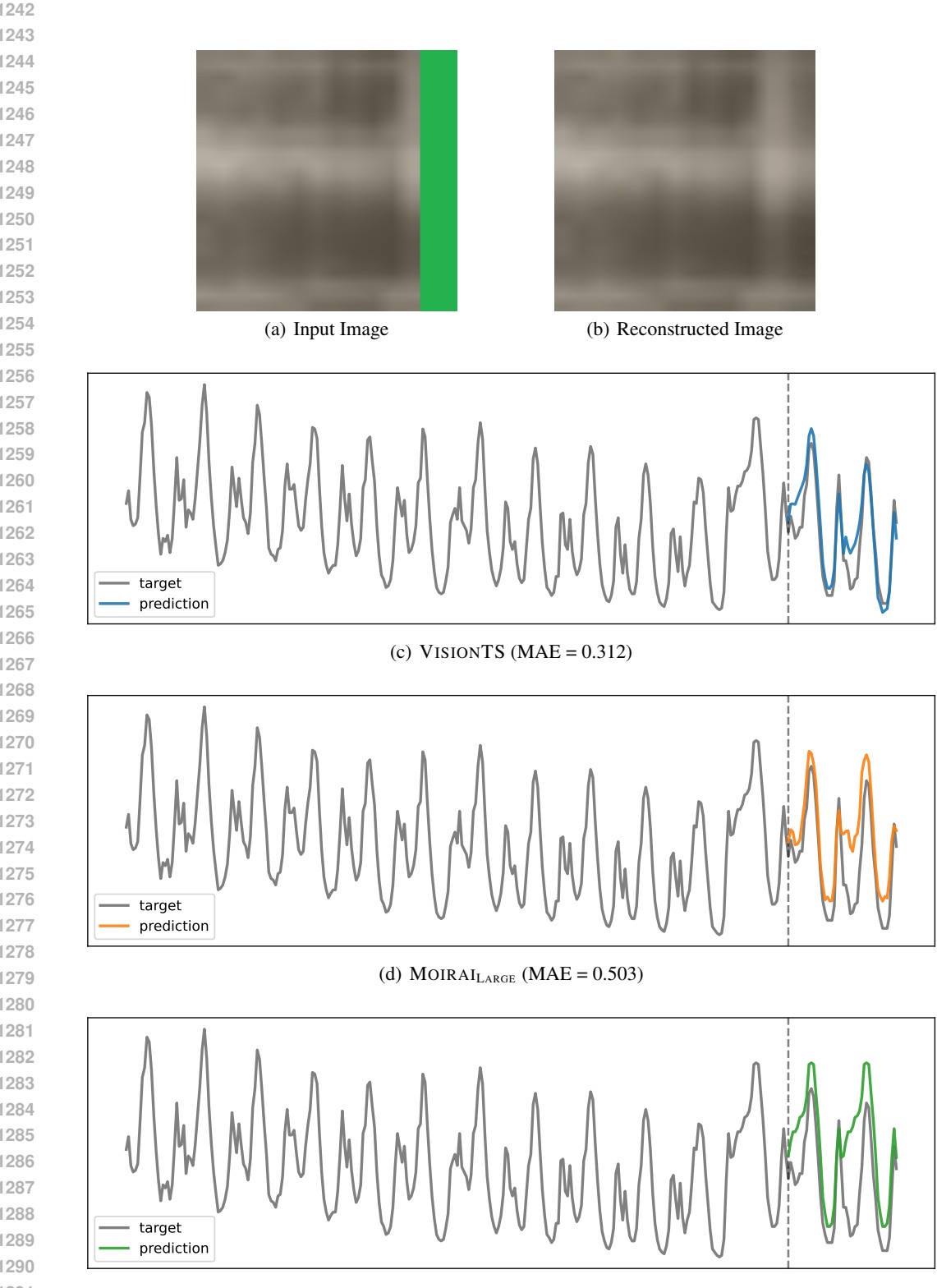

(a) Input Image      (b) Reconstructed Image

(c) VISIONTS (MAE = 0.312)

(d) MOIRAI$_{\text{LARGE}}$ (MAE = 0.503)

(e) Seasonal Naïve (MAE = 0.774)

Figure 10: Forecasting visualization on a sample from ETTh1. (a-b) Input/output images of VI-SIONTS. (c-e) Forecasting visualization.

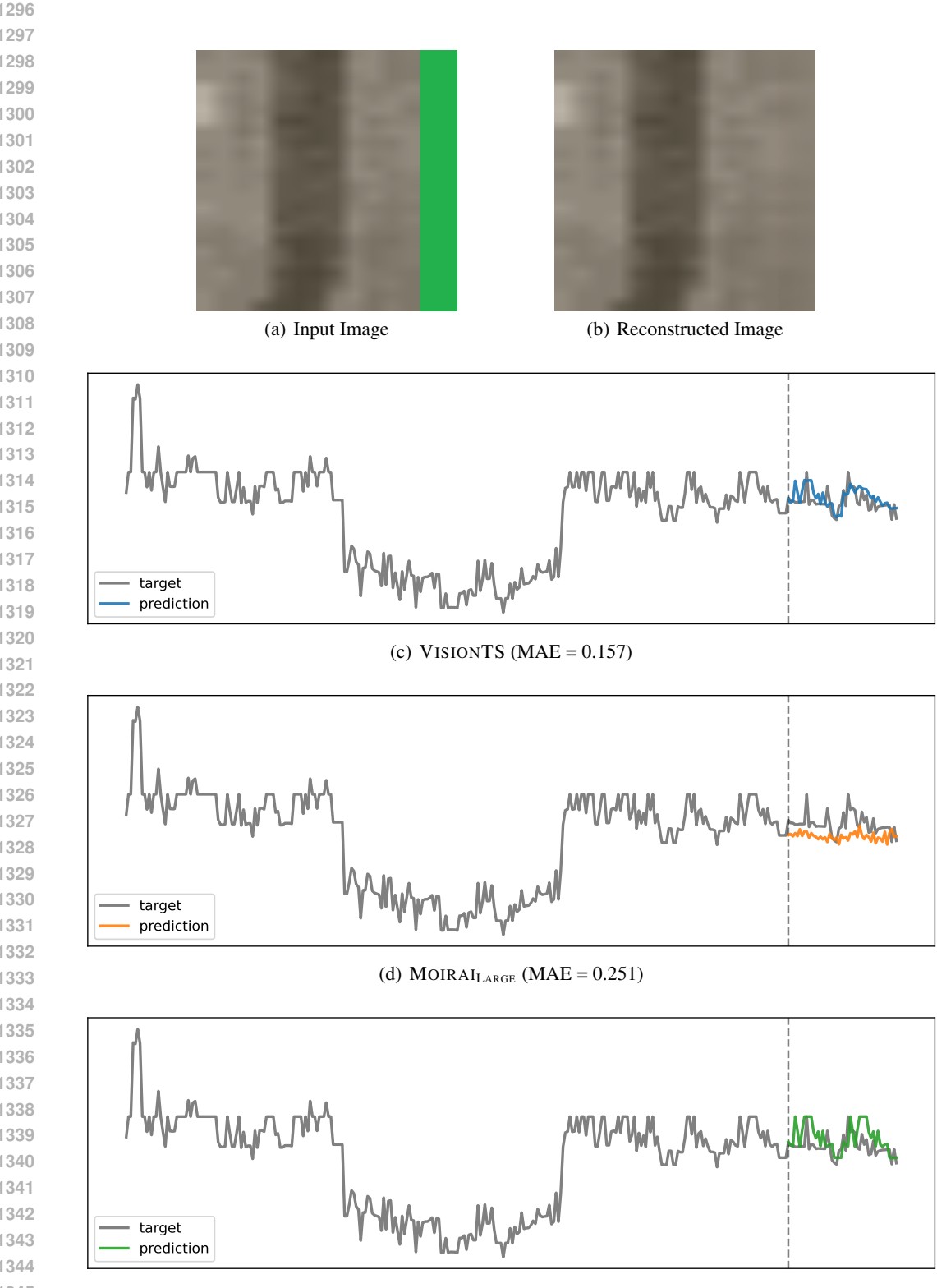

(a) Input Image       (b) Reconstructed Image

(c) VISIONTS (MAE = 0.157)

(d) MOIRAI_LARGE (MAE = 0.251)

(e) Seasonal Naïve (MAE = 0.235)

Figure 11: Forecasting visualization on a sample from ETTh2. (a-b) Input/output images of VI-SIONTS. (c-e) Forecasting visualization.

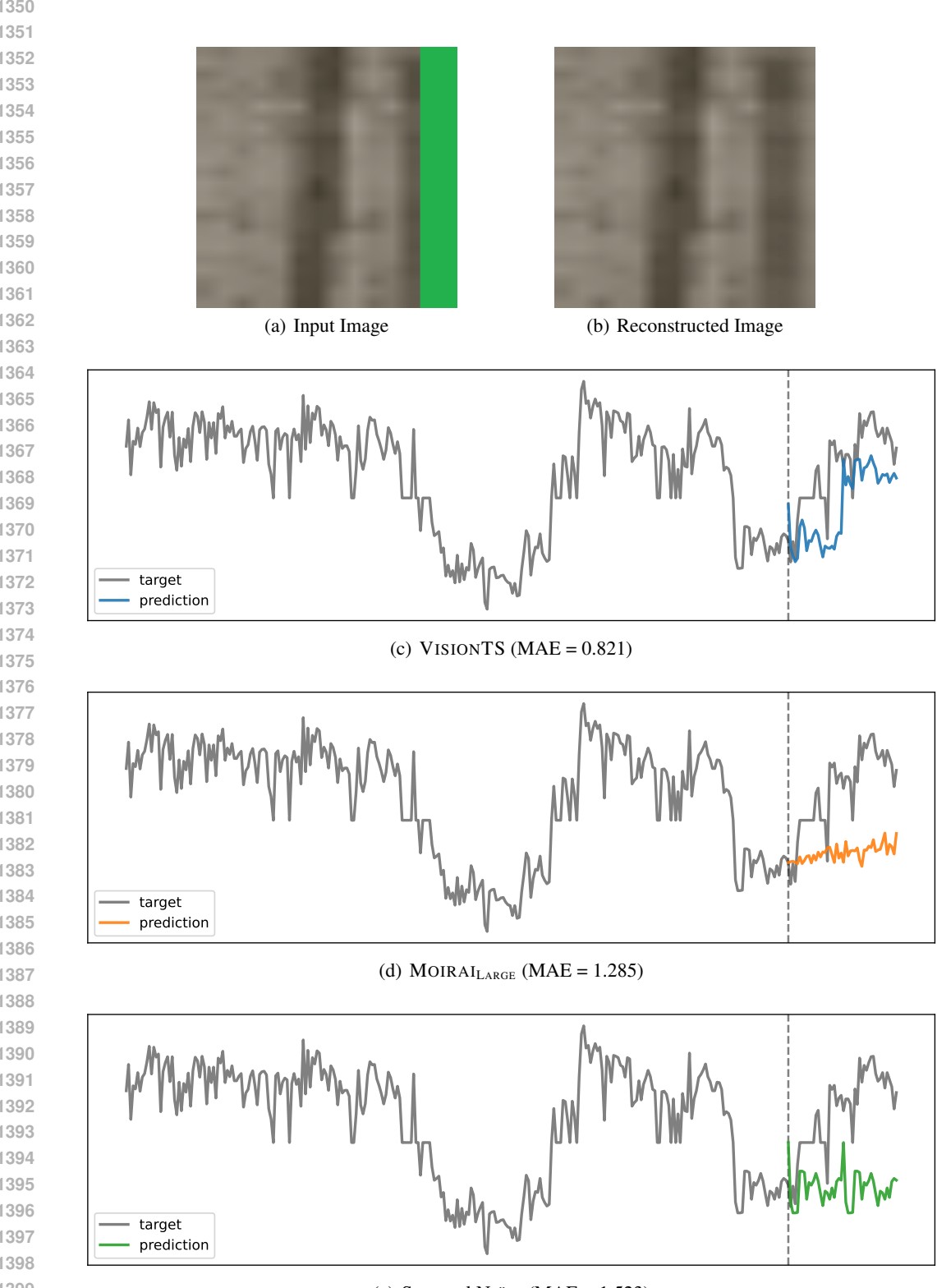

(a) Input Image        (b) Reconstructed Image

(c) VISIONTS (MAE = 0.821)

(d) MOIRAI$_{\text{LARGE}}$ (MAE = 1.285)

(e) Seasonal Naïve (MAE = 1.523)

Figure 12: Forecasting visualization on a sample from ETTh2. (a-b) Input/output images of VI-SIONTS. (c-e) Forecasting visualization.

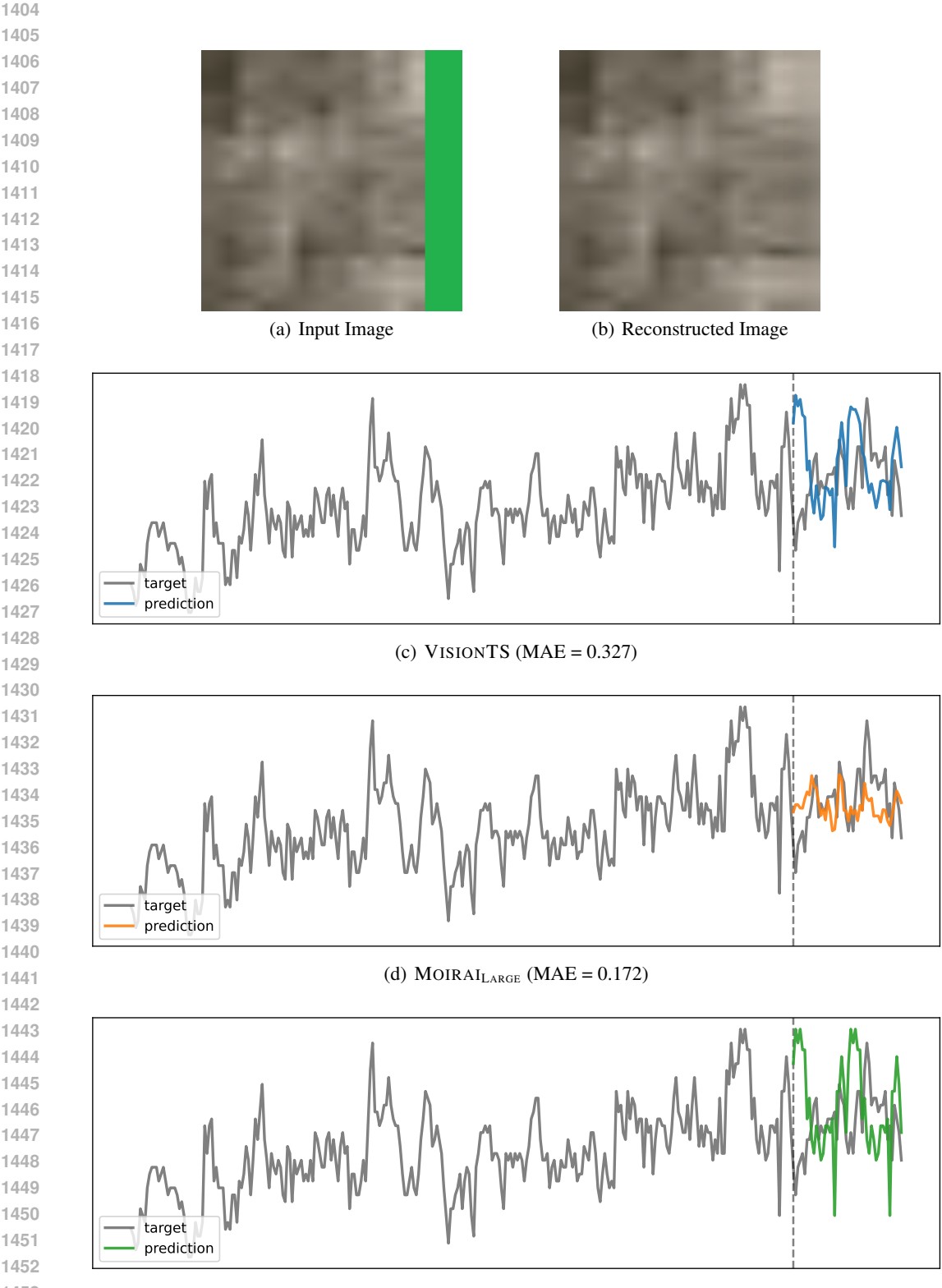

(a) Input Image  (b) Reconstructed Image

(c) VISIONTS (MAE = 0.327)

(d) MOIRAI_LARGE (MAE = 0.172)

(e) Seasonal Naïve (MAE = 0.364)

Figure 13: Forecasting visualization on a sample from ETTh1, where MOIRAI outperforms VISIONTS in terms of MAE. (a-b) Input/output images of VISIONTS. (c-e) Forecasting visualization.

