# OpenReview forum: "VisionTS: Visual Masked Autoencoders Are Free-Lunch Zero-Shot Time Series Forecasters"
_ICLR.cc/2025/Conference — Submitted to ICLR 2025_

### Official Review · Reviewer_Tfqf · 2024-11-02

**Soundness:** 2
**Presentation:** 1
**Contribution:** 2
**Rating:** 3
**Confidence:** 4

**Summary:**

This paper proposes using MAE, a masked autoencoder pretrained on 2D images with mask modeling pretraining, as a zero-shot forecasting model for time series data. The proposed method reformulates the time series forecasting task into a reconstruction task. Experimental results show that this method can perform zero-shot forecasting and achieve state-of-the-art performance in most cases after fine-tuning.

**Strengths:**

- This paper provides an interesting view of using MAE pretrained on vision tasks as a zero-shot time series forecasting model.
- Results show that this model is better than text-based time series forecasting in zero-shot settings.
- An interesting approach is proposed to achieve this transformation from image to text.

**Weaknesses:**

# After reading the rebuttal:
I appreciate the authors' feedback on my concerns. However, as the responses still fail to resolve my concerns, I have decided to maintain my initial score.

- The statement, “VisionTS can work well for multivariate forecasting by using the channel independence strategy,” does not demonstrate that VisionTS effectively handles the relationships among multivariate data. While employing a channel independence strategy for multivariate handling is straightforward, it is not necessarily the optimal approach. Achieving good performance with this strategy might be more indicative of limitations in the benchmark rather than evidence that multivariate relationships are unnecessary.

- I appreciate that the authors provided some possible explanations for why MAE can be used for zero-shot learning on time series. However, none of these explanations serve as solid evidence. For instance, the motivation and case study could equally apply to other models, such as LLMs and time series. Additionally, the Modality Analysis and Figure 6 highlight the dissimilarity between images and most time series data, which contradicts the claim of “a relatively small gap between images and some time series.”

- The authors use LaMa as an example of another vision model. Given that LaMa employs a relatively weak vision backbone, I wonder whether using a vision model with stronger representation capabilities would result in better performance. Alternatively, carefully tuning the introduced hyperparameters might allow other models, such as LLMs or randomly initialized models, to achieve comparable performance.

- I appreciate the authors’ explanation regarding the tuning of hyperparameters. However, these results remain empirical and lack theoretical analysis.

# Initial review:
- Using MAE for forecasting only works for univariate forecasting, while the more challenging and important multivariate forecasting is not supported.
- It doesn't really make sense that MAE, which is pretrained solely on images, can perform reasonable zero-shot forecasting, as no time series information is encoded in the MAE pretraining. This work only transforms the format of time series data to make it compatible with MAE.
- The transformation between a univariate input X and the image-like 2D matrix is unclear. The explanation of how P is selected should be provided. How is P obtained from sampling frequency and Fast Fourier Transform? Why is this the right way to select P, or is P not important for the final performance?
- There are tricks and hyperparameters involved in aligning MAE with time series forecasting, but the explanations are unclear.
  - The standard deviation of the input is controlled by a hyperparameter r. While an initial explanation is given, more detailed experiments and justification should be provided for the choice of this value.
  - Why does setting c = 0.4 perform well, aside from the experimental results?
- Given the many tricks and unclear explanations, I doubt whether MAE can truly perform zero-shot time series forecasting or if it merely overfits to these time series datasets to achieve a favorable evaluation number.
- The inference cost comparison is only provided for these LLM-time series models, while a comparison with native time series models should also be included.

**Questions:**

- Why does MAE work for zero-shot learning on time series data?
- How do these tricks work, and why are these hyperparameters chosen aside from experimental results?-
- For more questions, please refer to the weakness sections.

---

> ### Author Response · Authors · 2024-11-15
> **Official Comment by Authors (1/2)**
>
> Thank you for your valuable comment. However, we kindly note that **many of your concerns are already addressed in our original manuscript**. Below are our responses to your concerns:
>
> > W1: only works for univariate forecasting
>
> - We argue that **VisionTS can work well for multivariate forecasting by using the channel independence strategy.** Many related works [1,2,3,4,5] proves that this strategy is effective and even performs better than vanilla multivariate models. Importantly, our long-term TSF benchmarks are all multivariate datasets. As shown in Table 1, VisionTS achieves highly competitive zero-shot results on this multivariate dataset.
> - For future work, one can explore encoding multivariate inputs as multichannel images for fine-tuning the MAE model, as mentioned in Section 7. However, this is beyond the scope of the current paper, which focuses on zero-shot forecasting.
>
> > W2: It doesn't make sense that MAE can perform reasonable zero-shot forecasting.
> >
> > W5: I doubt whether MAE can truly perform zero-shot time series forecasting or if it merely overfits these time series datasets.
> >
> > Q1: Why does MAE work for zero-shot learning on time series data?
>
> We agree that understanding why MAE performs well is a key question and we have tried our best to answer it (dedicated almost **one page** in our paper). Here's what we explored:
>
> 1. **Sufficient Motivation**: In Introduction, we regard the pixel fluctuations of an image as a sequence and enumerate four similarities between this sequence and real-world time series: similar modality, origin, information density, and features.
> 2. **Case Study**: In Figure 2, we present an ImageNet image that displays many time series-related features, which qualitatively demonstrates that vision models trained on ImageNet can understand time series. This finding was approved by `Reviewer BBkW`.
> 3. **Theoretical Study**: In Section 4.2 (Modality Analysis), we analyze the similarities between ImageNet and time series representations. This provides a solid foundation for the proposed method and has been acknowledged by `Reviewer 3mCw`.
> 4. **Extensive Empirical Study**: To rule out the case that VisionTS is tailored only to a few datasets, we tested VisionTS across 43 datasets from various domains, far surpassing many existing TSF papers [1,2,3,4,5] which typically focus on fewer than 8 long-term TSF benchmarks. **We argue that VisionTS does not "overfit datasets"**, since we keep both the model and the proposed hyperparameters (c and r) *frozen* for every dataset.
> 5. **Beyond MAE**: We also tested LaMa [6], another visual inpainting model, and found that it still achieves strong zero-shot performance (See our `global response` for details). This suggests that the success comes from the similarity between images and time series, not just the MAE model.
>
> **As the first paper to apply vision models to zero-shot TSF, we understand your doubts and concerns.** We hope the above explanations can address them and inspire future work to explore deeper causes. If you have differing views, we welcome your further supporting evidence.
>
> > W3: The explanation of how P is selected should be provided. How is P obtained from sampling frequency and Fast Fourier Transform? Why is this the right way to select P, or is P not important for the final performance?
>
> - We would like to clarify that **we have explained how P is selected in Appendix A.2** (referenced in Line 194). In brief, we first determine a range of P based on sampling frequency and select the optimal P on the validation set. This frequency-based strategy is also employed in [7].
> - To demonstrate that P affects performance, the table below compares MSE results (averaged across four prediction lengths) with P=1 versus the original VisionTS, highlighting the importance of choosing the appropriate P.
>
> |  | VisionTS | VisionTS (P=1) |
> | ----- | -------- | -------------- |
> | ETTh1 | 0.390 | 0.840 |
> | ETTh2 | 0.333 | 0.424  |
> | ETTm1 | 0.374  | 0.660 |
> | ETTm2 | 0.282  | 0.312  |
>
> > W4: The standard deviation of the input is controlled by a hyperparameter r. While an initial explanation is given, more detailed experiments and justification should be provided for the choice of this value. Why does setting c = 0.4 perform well, aside from the experimental results?
> >
> > Q2: How do these tricks work, and why are these hyperparameters chosen aside from experimental results?
>
> - **We have conducted detailed experiments to verify our choice of hyperparameters**. Figures 7 and 8 (referenced in Line 370) show that the optimal values for both r and c are around 0.4. This value (0.4) worked well and reduced the need for further tuning on hyperparameters.
> - Additionally, **we have explained our choice of relatively low hyperparameters in the main text.** The low r prevents the forecasts from exceeding the magnitude (Line 210), and the low c ensures the number of visible patches aligns with MAE pretraining (Line 230).
>
>
>
> ---
>
> To be continued.

---

> ### Author Response · Authors · 2024-11-15
> **Official Comment by Authors (2/2)**
>
> > W6: An inference cost comparison with native time series models should also be included.
>
> Thank you for the suggestion. We tested the runtime of PatchTST and DeepAR, both native time series models without pretraining, as follows.
>
> | Context len  | 1000     | 2000     | 3000     | 4000     | 1000     | 1000     | 1000     | 1000     |
> | ------------ | -------- | -------- | -------- | -------- | -------- | -------- | -------- | -------- |
> | **Pred len** | **1000** | **1000** | **1000** | **1000** | **1000** | **2000** | **3000** | **4000** |
> | PatchTST     | 0.01     | 0.01     | 0.01     | 0.01     | 0.01     | 0.02     | 0.03     | 0.04     |
> | DeepAR       | 0.26     | 0.32     | 0.37     | 0.43     | 0.26     | 4.06     | 6.1      | 8.17     |
> | **VisionTS** | 0.04     | 0.03     | 0.03     | 0.03     | 0.04     | 0.04     | 0.05     | 0.05     |
>
> If the above explanations do not fully address your concerns, we welcome your further insights. **Thanks again for your constructive comment and we look forward to your feedback.**
>
> *References*
>
> [1] Multi-patch prediction: Adapting LLMs for time series representation learning, ICML 2024.
>
> [2] A time series is worth 64 words: Long-term forecasting with transformers, ICLR 2022.
>
> [3] Time-LLM: Time series forecasting by reprogramming large language models. ICLR 2024.
>
> [4] One fits all: Power general time series analysis by pretrained lm. NeurIPS 2023.
>
> [5] SparseTSF: Modeling long-term time series forecasting with 1k parameters. ICML 2024.
>
> [6] LaMa: Resolution-robust Large Mask Inpainting with Fourier Convolutions. WACV 2022.
>
> [7] GluonTS: Probabilistic and Neural Time Series Modeling in Python. JMLR 2020.

---

> ### Author Response · Authors · 2024-11-24
> **Kindly Reminder on Deadline and Further Clarification on MAE's Effectiveness**
>
> Dear Reviewer Tfqf,
>
> Thank you again for your thoughtful review! We fully understand your skepticism about MAE's effectiveness for time series — indeed, we were also surprised during our early research! To address your concerns, we have conducted extensive analyses such as modality similarity, which have been positively received by other reviewers, e.g., `a solid foundation for the proposed method` (3mCw) and `reasonable motivations in Fig.2` (BBkW).
>
> We want to further address your concerns *from a baseline perspective*. While MAE pre-training does not include time series data (as you noted!), the zero-shot time series foundation models we compared against (e.g., Moirai, TimesFM, LLMTime) **also does *not* include domain-specific time series**. These models explore **cross-domain** transferability, but the heterogeneity of time series patterns across domains (e.g., financial vs. weather) significantly limits their performance. In this context, it seems reasonable for a visual model to rival or exceed such cross-domain time series baselines, since images can also be regarded as a special "time series domain".
>
> Notably, we do *not* claim that zero-shot MAE surpasses full-shot forecasting methods. However, we show that MAE requires minimal fine-tuning (e.g., tuning LN for one epoch) to acquire time series information, demonstrating its strong transferability from images to time series.
>
> We also want to note that many of your concerns regarding method explanations and details have been addressed in our initial submission. Other reviewers have acknowledged the clarity of our method, e.g., `clear explanations of the methodology` (3mCw) and `sufficient details about the proposed method` (BBkW).
>
> **We are willing to share our exciting cross-modality findings with the TSF community and believe it will inspire more great ideas. While we understand that new findings often spark debate, we are concerned that your negative rating (3) might hinder us from sharing our interesting findings**. As the discussion *ends soon*, we would appreciate it if you could reevaluate our work based on our responses and contributions. Thank you for your time and consideration.
>
> Best regards,
>
> The Authors

---

> > ### Author Response · Authors · 2024-11-29
> > **Kindly Reminder**
> >
> > Dear Reviewer,
> >
> > Thank you again for your time and feedback on our submission. We would like to kindly remind you that it has been **two weeks** since our initial response. Given that some of your comments may have stemmed from misunderstandings — such as overlooking key points in our paper or potential conflicting with other reviewers’ opinions — we believe further discussion would improve both the review process and the quality of our paper. Your continued engagement would be a valuable contribution to the ICLR community.
> >
> > Best,
> >
> > The Authors

---

> ### Author Response · Authors · 2024-12-04
> **Reply to "After reading the rebuttal:" (1/2)**
>
> # Reply to "After reading the rebuttal:"
>
> Thank you for your response to our reply in the review content. **However, we believe it would have been better if you had responded to us during the discussion phase**, as we would have received email notifications in a timely manner. Below are our responses to your further concerns:
>
> > R1: The statement, “VisionTS can work well for multivariate forecasting by using the channel independence strategy,” does not demonstrate that VisionTS effectively handles the relationships among multivariate data. While employing a channel independence strategy for multivariate handling is straightforward, it is not necessarily the optimal approach.
>
> We would like to emphasize that our core contribution is *not* to investigate how to capture the multivariate dependency, but rather to explore using vision models for TSF. As such, we adopted the channel independence strategy, **which is quite common in the current time series research**. We agree that it is not necessarily the optimal approach and plan to explore multivariate dependencies as future work. Considering the substantial content in our paper, incorporating such investigation in a single paper will make it hard to understand our key contribution (i.e., zero-shot forecasting).
>
> > R1: Achieving good performance with this strategy might be more indicative of limitations in the benchmark rather than evidence that multivariate relationships are unnecessary.
>
> We want to emphasize that we have already used a sufficient number of datasets (a total of **43**, including LTSF, monash, and PF). This is far larger than many related papers, which typically focus on fewer than 8 long-term TSF benchmarks or a single domain (e.g., energy). **If you believe there are issues with the benchmarks, could you please provide new benchmarks for us to test?** Once again, we regret that this concern was raised now rather than during the discussion phase, as we could have shared the experimental results for the new benchmarks you can provide.
>
> > R2: However, none of these explanations serve as solid evidence. For instance, the motivation and case study could equally apply to other models, such as LLMs and time series.
>
> We respectfully argue that these motivations and case studies can not be generalized to LLMs. For example, images are continuous, while the language used in LLMs is discrete. Furthermore, language seldom displays features shown in Figure 2. We agree that they can be applied to other time series  models, and thus it requires further **modality analysis**. See below:
>
> > R2: Additionally, the Modality Analysis and Figure 6 highlight the dissimilarity between images and most time series data, which contradicts the claim of “a relatively small gap between images and some time series.”
>
> We would like to highlight that Figure 6 do not contradict such claims, as we emphasize that this applies to *some* time series, rather than *all* time series. This also explains why VisionTS performs well on some datasets, but not all the datasets. Additionally, Figure 6 demonstrates that most real-world time series exhibit significant heterogeneity, even more than the gap between images and time series. Importantly, images can serve as a bridge to connect them. **This further explains why vision models perform better than some cross-domain time series models.**
>
> As the first work to apply vision models to zero-shot TSF, we believe our explanation is already quite comprehensive. **If you have suggestions on how to provide more solid evidence, we would greatly appreciate it.**
>
> ---
>
> (to be continued)

---

> ### Author Response · Authors · 2024-12-04
> **Reply to "After reading the rebuttal:" (2/2)**
>
> # Reply to "After reading the rebuttal:" (part 2)
>
> > R3: The authors use LaMa as an example of another vision model. Given that LaMa employs a relatively weak vision backbone, I wonder whether using a vision model with stronger representation capabilities would result in better performance.
>
> In Section 4.2 (Scaling Analysis), we have already demonstrated that using more complex vision models may lead to a slight decrease in time series forecasting performance. The reason is that more powerful vision models might overfit certain image-related, yet time series-irrelevant, nuanced features. Exploring whether there exists a scaling law for vision models in time series forecasting is undoubtedly a promising avenue for future development.
>
> > R3: Alternatively, carefully tuning the introduced hyperparameters might allow other models, such as LLMs or randomly initialized models, to achieve comparable performance.
>
> **We would like to emphasize that the hyperparameters c and r we proposed are unique to vision models and cannot be applied to other models.** The introduction of these two specific hyperparameters is to bridge the gap between visual MAE and downstream time series tasks. Additionally, we kindly remind you that a randomly initialized model cannot perform **zero-shot** forecasting, regardless of the additional hyperparameters introduced.
>
> > R4: I appreciate the authors’ explanation regarding the tuning of hyperparameters. However, these results remain empirical and lack theoretical analysis.
>
> We have provided a qualitative analysis for the selection of hyperparameters. As far as we know, most papers lack a theoretical analysis that can directly compute the optimal hyperparameters such as learning rate, model size, and batch size. Do you have any insight on how to calculate our optimal hyperparameters through theoretical analysis?
>
> **We hope our responses can help bridge the gap between the reviewers' and the authors' academic perspectives, working together toward a new direction of time series forecasting. Regardless of whether this paper is accepted, we sincerely appreciate your valuable feedback.**

---

### Official Review · Reviewer_BBkW · 2024-11-03

**Soundness:** 3
**Presentation:** 3
**Contribution:** 3
**Rating:** 8
**Confidence:** 3

**Summary:**

This paper provides a new perspective that treating time series forecasting problems as image reconstruction problems. And directly take the pretrained MAE model to conduct such tasks. Specifically, the paper just performs segmentation, normalization and alignment to transform 1D time series data into 2D data, and predict masked tokens for TSF tasks. The results shows that this algorithm has decent performance under both zero-shot and full-shot settings.

**Strengths:**

1. This paper provides a novel view for TSF problems, and explain with reasonable motivations in Fig.2.
2. The method part provides sufficient details about the proposed method, making it easy to read and follow.
3. This paper provides experiments on a wide range of tasks to validate the proposed method.

**Weaknesses:**

1. In the alignment step, the scaling process is influenced by both the context length $L$ and the prediction length $H$. It seems that the forecasted result at $L+1$ may change due to whether we will predict $L+2$ or not, which is counter-intuitive.

Since the method map any-length series to a fixed resolution, I suppose it requires the model to understand patterns of different frequency. This might be hard for a pretrained vision model. It is trained only with natural images, which cannot cover drastic changes such as noises.

2. If the similarity between image and TSF holds true, maybe the authors should try other image inpainting models for the TSF tasks. We may assume that thoses models can have a better performance for image reconstruction.

3. In full-shot experiments, the authors decide to tune all the LN parameters. It is uncommon for the validation in vision pretrain model. Usually we either finetune the full model or only the last linear layer (e.g. linear probe). I wonder the reason that the authors prefer the LN tuning.

**Questions:**

My main concerns are listed in weaknesses. Regarding these comments, I would like the authors to check if MAE is suitable for TSF.

---

> ### Author Response · Authors · 2024-11-15
>
> Thank you for your positive feedback and valuable comments on our paper. Below are our responses to your concerns:
>
> > W1 (1): The scaling process is influenced by both the context length and prediction length ... which is counter-intuitive.
>
> - We agree that changing the prediction length affects performance. However, we would like to clarify that **this is also a feature of many TSF papers**. Traditional TSF models [1,2,3] require separate models for each prediction length. A long-term forecasting model typically performs differently on short-term predictions compared to a model specifically trained for short-term forecasting.
>
> > W1 (2): Since the method map any-length series to a fixed resolution, I suppose it requires the model to understand patterns of different frequency. This might be hard for a pretrained vision model.
>
> - This is an interesting point. We believe pre-trained vision models can still capture patterns at different scales. MAE pre-training includes random cropping (RandomResizedCrop) as data augmentation, which helps the model recognize patterns at various scales for a fixed resolution. Therefore, even with fixed-resolution inputs, VisionTS can capture multi-scale patterns in time series data.
>
> > W2: Should try other image inpainting models.
> >
> > Q1: I would like the authors to check if MAE is suitable for TSF.
>
> - Thank you for the suggestion. We tested another inpainting model, LaMa [4], and the results are discussed in our `global response`. VisionTS with LaMa performs similarly to Moirai (Small/Large) in zero-shot settings and outperforms other few-shot baselines. However, its performance is slightly lower than MAE’s. This may be due to LaMa overfitting image-specific features during training, as noted in Section 4.2 (Scaling Analysis). We believe a more balanced Visual MAE is better suited for TSF.
>
> > W3: Why tuning on LN parameters
>
> We followed the approach from [5], which fine-tuned only the LN parameters of the LLM for the TSF task. To verify this choice, we have compared other tuning strategies, such as full tuning, tuning MLP, attention, or bias components. As shown in Table 8, tuning the LN parameters leads to the best performance gains.
>
> **We hope these clarifications address your concerns. We look forward to your feedback.**
>
> *References*
>
> [1] A time series is worth 64 words: Long-term forecasting with transformers, ICLR 2022.
>
> [2] Are transformers effective for time series forecasting? AAAI 2023.
>
> [3] Timesnet: Temporal 2d-variation modeling for general time series analysis. ICLR 2023.
>
> [4] LaMa: Resolution-robust Large Mask Inpainting with Fourier Convolutions. WACV 2022.
>
> [5] One fits all: Power general time series analysis by pretrained lm. NeurIPS 2023.

---

> ### Author Response · Authors · 2024-11-24
> **Kindly Reminder on Deadline**
>
> Dear Reviewer BBkW,
>
> Thank you again for your thoughtful review! We greatly appreciate your positive feedback and find your comments (e.g., `resolution issue` and `another inpainting model`) very insightful. We would be grateful for your thoughts on our previous response to help improve the quality of our paper. **As the discussion ends soon, we truly value your perspective and kindly request your support**. Thank you for your time and consideration.
>
> Best regards,
>
> The Authors

---

> > ### Comment · Reviewer_BBkW · 2024-11-26
> > **Thanks for feedback**
> >
> > Thank the authors for feedbacks, especially for the image inpainting experiments. Other reviewers point out that the underlying reason why pretrained MAE can solve TSF tasks is unclear, and this method involves hyperparameters to tune. Nonetheless, this paper provides an interesting perspective and experiment results, and I keep a positive attitute towards this paper.

---

> ### Author Response · Authors · 2024-11-26
> **Thanks for your positive attitute and we would like to clarify the misunderstanding**
>
> Thank you for your positive attitude and response! However, we would like to respectfully clarify a misunderstanding you (and potentially other reviewers) may have:
>
> - We’ve conducted extensive analyses (dedicated almost one page in our paper) to understand why the pretrained MAE could work well. Following your suggestion, we also added inpainting experiments, which further indicate that its success comes from the similarity between images and time series, not just the MAE model itself. If you have any suggestions for further exploring the underlying reasons, we’d be happy to implement them!
>
> - Our method’s hyperparameters do not require tuning. In all main experiments, we used a default setting of  $c=r=0.4$, which already performs strongly across all the domains. See Figures 7 and 8 for our hyperparameter analysis.
>
> We appreciate your insights and would welcome further discussion. As a friendly reminder, according to the latest notification from ICLR, the discussion deadline has been extended by one week.

---

### Official Review · Reviewer_3mCw · 2024-11-04

**Soundness:** 2
**Presentation:** 2
**Contribution:** 2
**Rating:** 5
**Confidence:** 4

**Summary:**

The paper titled "VISIONTS: VISUAL MASKED AUTOENCODERS ARE FREE-LUNCH ZERO-SHOT TIME SERIES FORECASTERS" introduces a approach to time series forecasting (TSF) by leveraging pre-trained visual masked autoencoders (MAEs) on natural images. The key idea is that the pixel variations in images can be interpreted as temporal sequences, which share intrinsic similarities with time series data. The authors reformulate TSF as an image reconstruction task, allowing the pre-trained MAE to perform zero-shot forecasting without further adaptation. Extensive experiments demonstrate that VISIONTS achieves superior zero-shot forecasting performance compared to existing TSF foundation models and can further improve with minimal fine-tuning. The findings suggest that visual models may offer a "free lunch" for TSF and highlight the potential for future cross-modality research.

**Strengths:**

1. The authors provide extensive experimental results across 43 TSF benchmarks, demonstrating VISIONTS's effectiveness in zero-shot and few-shot settings.

2. The paper offers a theoretical analysis of the similarities between images and time series, providing a solid foundation for the proposed method. This contributes to a deeper understanding of cross-modality transferability.

3. The paper is well-written, with clear explanations of the methodology, experiments, and results. The figures and tables are informative and support the narrative effectively.

**Weaknesses:**

1. The current approach is limited to univariate forecasting, which may restrict its applicability in real-world scenarios where multivariate time series are common. This limitation could be a barrier to broader adoption.

2. While the use of pre-trained models is a strength, it also introduces a dependency on external datasets for performance, which may not always be ideal or ethically sound. Did you use EVA-CLIP?

3. The paper does not fully address how VISIONTS scales with larger and more complex time series datasets, which is a critical consideration for practical applications.

**Questions:**

Please refer to weaknesses.

---

> ### Author Response · Authors · 2024-11-15
>
> Thank you for your valuable comment. We kindly note that **most of your concerns are common in many related works published in top conferences**. Please find our responses below:
>
> > W1. limited to univariate forecasting
>
> - We argue that **VisionTS can still work well for multivariate forecasting by using the channel independence strategy.** Many related works [1,2,3,4,5] prove that this strategy is effective and even performs better than vanilla multivariate models. Importantly, our long-term TSF benchmarks are all multivariate datasets. As shown in Table 1, VisionTS achieves highly competitive zero-shot results on this multivariate dataset.
> - For future work, one can explore encoding multivariate inputs as multichannel images for fine-tuning the MAE model, as mentioned in Section 7. However, this is beyond the scope of the current paper, which focuses on zero-shot forecasting.
>
> > W2 (1) A dependency on external datasets
>
> - We would like to clarify that using external datasets to pre-train TSF models (i.e., foundation models) is a promising direction in the related works [3,4,7,8] and the focus of our paper. All of the baselines in our paper require external or internal time series datasets. Compared to them, VisionTS leverages "free-lunch" pre-trained vision models. This allows us to not only **avoid the need for *external* time series datasets**, but also achieve strong zero-shot performance **without training on *internal* time series datasets**.
>
> > W2 (2) Did you use EVA-CLIP?
>
> - EVA-CLIP uses additional training objectives (MIM and CLIP), making it not a pure MAE and unsuitable for VisionTS. Instead, we tested another vision model, LaMa [6]. Please refer to our global response for more details.
>
> > W3: scales with larger and more complex time series datasets
>
> - We followed [7] and tested ***43*** time series datasets, which cover a wide range of domains, data scales, and temporal patterns, making them highly heterogeneous and complex. **This is far larger than many related papers [1,2,3,4,5], which typically focus on fewer than *8* long-term TSF benchmarks or single domain (e.g., energy)**. Therefore, we respectfully argue that it is unfair to criticize our scaling efforts.
>
> We hope these clarifications address your concerns. **Thanks again for your valuable comment and we look forward to your feedback.**
>
> *References*
>
> [1] Multi-patch prediction: Adapting LLMs for time series representation learning, ICML 2024.
>
> [2] A time series is worth 64 words: Long-term forecasting with transformers, ICLR 2022.
>
> [3] Time-llm: Time series forecasting by reprogramming large language models. ICLR 2024.
>
> [4] One fits all: Power general time series analysis by pretrained lm. NeurIPS 2023.
>
> [5] SparseTSF: Modeling long-term time series forecasting with 1k parameters. ICML 2024.
>
> [6] LaMa: Resolution-robust Large Mask Inpainting with Fourier Convolutions. WACV 2022.
>
> [7] Unified training of universal time series forecasting transformers. ICML 2024.
>
> [8] A decoder-only foundation model for time-series forecasting. ICML 2024.

---

> ### Author Response · Authors · 2024-11-24
> **Kindly Reminder on Deadline**
>
> Dear Reviewer 3mCw,
>
> Thank you again for your thoughtful review! As stated in the previous response, we understand your concerns regarding `channel independence forecasting` (W1) and `external datasets` (W2), but we respectfully remind you that these are promising research directions in the recent work. Additionally, regarding `dataset scaling` (W3), our work even goes beyond many existing studies.
>
> With this in mind, we would appreciate it if you could reconsider your negative score (5), as **it could also lead to rejecting other related top-tier TSF papers and may not fully reflect the contributions of our work**. As you noted, our contribution advances `a deeper understanding of cross-modality transferability` (S2).
>
> **As the discussion ends soon**, we truly value your perspective and hope you’ll take another look. Thank you for your time and consideration.
>
> Sincerely,
>
> The Authors

---

> ### Author Response · Authors · 2024-11-29
> **Kindly Reminder**
>
> Dear Reviewer,
>
> Thank you again for your time and feedback on our submission. We would like to kindly remind you that it has been **two weeks** since our initial response. Given that some of your comments may lack specificity on our paper and seem to generalize to many related works, we believe further discussion would improve both the review process and the quality of our paper. Your continued engagement would be a valuable contribution to the ICLR community.
>
> Best,
>
> The Authors

---

### Author Response · Authors · 2024-11-15
**Global Comments**

## Global comment

We thank the reviewers for their insightful comments and constructive feedback. We're pleased that they agree our paper offers a `novel and interesting perspective` (R BBkW, R Tfqf). The paper is recognized for its `strong motivation` (R BBkW), `solid theoretical foundation` (R 3mCw), and `extensive experiments` (R 3mCw, R BBkW). The reviewers also appreciate the paper’s `clarity and detail` (R 3mCw, R BBkW). **We believe that our work opens a new road to building TSF foundation models and can inspire future cross-modality research.** We value their suggestions and will incorporate all feedback into the final version.

Our proposed VisionTS is initially based on a visual MAE. To explore its potential with other vision models, we also tested LaMa [1], a visual inpainting model. The MSE results (averaged across four prediction lengths) are as follows.

|                               | **ETTh1** | **ETTh2** | **ETTm1** | **ETTm2** | **Average** |
| ----------------------------- | --------- | --------- | --------- | --------- | ----------- |
| ***Traditional baselines***   |           |           |           |           |             |
| ARIMA                         | 0.912     | 0.516     | 0.707     | 0.405     | 0.635       |
| Seasonal Naive                | 0.600     | 0.483     | 0.489     | 0.358     | 0.482       |
| ***Few-shot methods***        |           |           |           |           |             |
| TimeLLM                       | 0.556     | 0.370     | 0.404     | 0.277     | 0.402       |
| GPT4TS                        | 0.590     | 0.397     | 0.464     | 0.293     | 0.436       |
| ***Zero-shot methods***       |           |           |           |           |             |
| Moirai (Small)                | 0.400     | 0.341     | 0.448     | 0.300     | 0.372       |
| Moirai (Large)                | 0.510     | 0.354     | 0.390     | **0.276** | 0.382       |
| **VisionTS (backbone: MAE)**  | **0.390** | **0.333** | **0.374** | 0.282     | **0.345**   |
| **VisionTS (backbone: LaMa)** | 0.425     | 0.376     | 0.400     | 0.294     | 0.374       |

Importantly, **all baselines, except VisionTS, were trained on time series datasets**. The results show that VisionTS with LaMa performs similarly to Moirai (Small/Large) in the zero-shot setting, and outperforms other few-shot models and traditional baselines. This suggests that the performance is driven by the inherent similarity between images and time series, not solely by the MAE model.

We also found that LaMa's performance lags behind MAE's. A possible reason is that LaMa may have overfitted image-specific nuanced features during training, as discussed in Section 4.2 (Scaling Analysis).

Next, we will address each reviewer's concerns individually.

*References*

[1] LaMa: Resolution-robust Large Mask Inpainting with Fourier Convolutions. WACV 2022.

---

> ### Author Response · Authors · 2024-11-17
> **Paper has been revised and we are eager to hear your further valuable feedback**
>
> Dear Reviewers:
>
>
> Thank you for your diligent review of the VisionTS paper! We have uploaded the latest revised version, with $\textcolor{purple}{\textbf{purple}}$ markings indicating new or modified content. **We are eager to hear your further valuable feedback.**
>
> Here is a quick summary of the revisions:
>
> - We added experimental results using LaMa as a new backbone for VisionTS, to demonstrate that VisionTS's performance is not solely driven by MAE, but rather by the intrinsic similarity between images and time series (credit to `Reviewer BBkW`).
> - We included a runtime comparison of native TSF models and reported the performance of changing P (credit to `Reviewer Tfqf`).
> - We revised several wordings to emphasize the effectiveness of the channel independence strategy (credit to `Reviewer 3mCw` and `Tfqf`) and ensured that the details in the appendix (e.g., hyperparameter analysis) are sufficiently referenced in the main text (credit to `Reviewer Tfqf`).
>
> Best regards,
>
> Authors.

---

### Meta-Review · Area_Chair_xjFN · 2024-12-20

**Metareview:**

This paper proposes VisionTS, an approach that reveals the intrinsic similarities between time series and images, reformulates time series forecasting into a masked image modeling task and employs ImageNet-pretrained masked autoencoder (MAE) for zero-shot and full-shot forecasting to deliver superior performance.

This paper provide a very interesting and novel perspective on time-series forecasting. The technical insight is well above the bar of ICLR.

However, both Reviewer Tfqf and Reviewer 3mCw raised concerns that VisionTS is limited to univariate forecasting, especially in its highlighted zero-shot setting. While the ignorance of multivariate relationships is the "limitations in the benchmark", the strong experimental results only on such limited benchmarks indeed weaken the claim of the intrinsic connection between time series and images.

**Additional Comments On Reviewer Discussion:**

Showing vision backbones can benefit time series forecasting is a great direction worth investigation. In such papers we can count down the novel ingredients proposed, but we do expect more thorough experiments which can "switch" our stereotypes and "rethink" this field. The paper can be significantly strengthened by providing more extensive empirical study, e.g. on multivariate forecasting and on different tasks.

---

### Decision · Program_Chairs · 2025-01-22

Reject